# Prequential Evidence Pruning: Information-Theoretic Edge Selection for Ordering-Based Causal Discovery

## Abstract

Ordering-based causal discovery reduces the complex problem of structure learning to parent selection given a candidate topological order. However, the pruning stage remains a critical bottleneck, as widely used procedures rely on marginal, additivity-constrained tests with manually tuned thresholds. These limitations often prevent the detection of non-additive interactions and hinder reproducibility. To address these challenges, we introduce *Prequential Evidence Pruning* (PEP), a framework that reformulates pruning as a local information-theoretic model selection problem. For each candidate edge, PEP computes the prequential log-evidence gain by evaluating the predictive density of a child node conditioned on its current co-parents using a sample-splitting strategy. An edge is retained if and only if this gain exceeds an adaptive Minimum Description Length (MDL) penalty that accounts for the sample size, the number of admissible parents, and the set size. Theoretically, we establish that the population target of the evidence gain corresponds to the Conditional Mutual Information (CMI). Furthermore, we prove that the statistic is stable under bounded log-loss regret and that prequential scoring provides finite-sample concentration guarantees. Empirically, instantiating PEP with a pre-trained tabular foundation model yields consistent improvements across diverse ordering backbones. Notably, our framework incorporates a hierarchical pruning strategy that enables scalability to higher-dimensional graphs, effectively elevating the pruning stage from marginal testing to scalable, context-aware evidence maximization.

## 1 Introduction

Causal discovery from observational data is fundamental to mechanistic understanding across science and engineering (Sachs et al., 2005; Van Koten & Gray, 2006; Hicks et al., 1980). However, exhaustive search over Directed Acyclic Graphs (DAGs) is super-exponential and therefore intractable without strong inductive biases (Bongers et al., 2021). Ordering-based methods address this computational challenge by first estimating a topological order and then pruning forward edges (Teyssier & Koller, 2012; Bühlmann et al., 2014; Peters et al., 2014; Rolland et al., 2022; Montagna et al., 2023c;b; Sanchez et al., 2023; Xu et al., 2024). While this two-stage paradigm has seen significant advances in the ordering step, the pruning step remains a practical bottleneck. Widely used procedures, such as those in Causal Additive Models (CAM) (Bühlmann et al., 2014), evaluate each candidate parent *marginally* under additivity constraints and make pruning decisions via fixed thresholds. This approach often obscures non-additive interactions among co-parents and induces unstable behavior across datasets. We illustrate this core challenge, which motivates our work, in Figure 1.

We propose *Prequential Evidence Pruning* (PEP), a principled framework that reformulates pruning as a localized cost–benefit analysis grounded in information theory. For a candidate edge $i \rightarrow j$ evaluated with its current co-parents $S \setminus \{i\}$, PEP quantifies a prequential log-evidence gain. This metric represents the improvement in the predictive log-likelihood of the child when conditioning on $X_i$ in addition to $X_{S \setminus \{i\}}$, computed using a sample-splitting strategy. Calculating evidence strictly out-of-sample mitigates in-sample optimism and ensures finite-sample stability. To convert this evidence into a robust decision, PEP compares the statistic $\delta_{i \rightarrow j}(q; S)$ against a computable Minimum Description Length (MDL) (Grünwald, 2007) penalty. This adaptive gate prices the

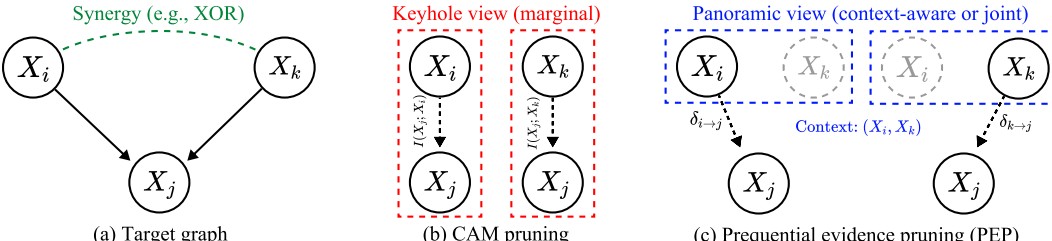

Figure 1: A conceptual illustration of our pruning framework. (a) The target graph depicts parents $X_i$ and $X_k$ having a synergistic effect on their child $X_j$. (b) In contrast, CAM pruning adopts a *keyhole view*, evaluating each parent in isolation. This approach fails to capture synergies when the marginal signal is null (e.g., $I(X_j; X_i) \approx 0$). (c) Our PEP framework addresses this limitation by adopting a *panoramic view*, which evaluates each parent ($X_i$) in the context of its co-parents ($X_k$) to compute an evidence gain ($\delta_{i \to j}$) that captures the interaction. For mathematical examples, see Appendix D.

combinatorial complexity of the search space given the topological order. Specifically, the per-sample threshold $\tau_j^{\mathrm{MDL}}(S, i)$ encodes the identity of the added parent among the admissible predecessors and the change in set cardinality (Eq. (2)–Eq. (3)). This yields an explicit, sample-size aware acceptance criterion that obviates the need for user-tuned significance levels.

Our framework is model-class agnostic and requires only a predictive component that outputs proper, calibrated conditional densities. In our experiments, we instantiate this component with a single pre-trained tabular foundation model (Hollmann et al., 2025b), which provides zero-shot, well-calibrated predictive densities for mixed data types. This allows our empirical study to focus on the contribution of the information-theoretic principle rather than on model-specific engineering.

**Contributions.** (1) We introduce a prequential, context-aware edge statistic that measures the out-of-sample predictive gain of a parent conditioned on its co-parents, effectively capturing synergistic and non-additive interactions. (2) We develop a decision gate based on the MDL principle, replacing user-tuned significance thresholds with a computed, adaptive penalty that enhances the robustness of pruning decisions. (3) We present a modular, plug-in pruning framework (PEP) that improves diverse ordering-based backbones by directly addressing their pruning shortcomings. (4) We provide theoretical guarantees for stability and introduce a Hierarchical Group Pruning strategy to address scalability. Extensive experiments demonstrate that PEP significantly outperforms state-of-the-art baselines on synthetic and real-world data, scaling effectively to higher-dimensional graphs.

## 2 RELATED WORK

**Ordering-based Causal Discovery.** Ordering-based approaches circumvent the super-exponential search over DAGs by first estimating a topological order and then pruning edges consistent with that order. Early works such as CAM (Bühlmann et al., 2014) and RESIT (Peters et al., 2014) pioneered this two-stage paradigm. A recent line of research, initiated by SCORE (Rolland et al., 2022), identifies leaves via properties of the score function and has given rise to several effective variants, including NoGAM (Montagna et al., 2023c), DAS (Montagna et al., 2023b), DiffAN (Sanchez et al., 2023), and CaPS (Xu et al., 2024). Despite significant progress in the ordering step, most pipelines still employ CAM-style, additivity-constrained post-processing for pruning. This approach evaluates candidates marginally and often fails to account for synergistic or non-additive interactions among parents. We address this under-explored bottleneck by introducing PEP. Our module performs joint, context-aware evaluation via a prequential log-evidence gain and utilizes a computed MDL penalty instead of tuned thresholds, allowing it to integrate with diverse ordering backbones without altering their ordering criteria. For additional related work in causal discovery, see Appendix B.5.

**Information-Theoretic Approaches in Causal Discovery.** Information theory has been foundational to causal discovery along two primary lines. Constraint-based procedures, such as the PC algorithm (Spirtes & Glymour, 1991), rely on statistical tests for conditional independence and use estimators of Conditional Mutual Information (CMI) with user-specified significance levels. In contrast,

score-based methods like GES (Chickering, 2002) optimize a global objective that balances model fit and complexity, often utilizing an MDL-derived penalty such as BIC. Our PEP framework synthesizes these two traditions. It uses an information-theoretic evidence statistic to quantify dependence in context and compares this against a computed MDL code-length penalty to make local edge decisions. This approach retains the semantic interpretability of CMI while inheriting the parsimony of MDL. Crucially, it avoids tuned thresholds and global parametric assumptions, making it applicable to nonparametric or amortized predictors (see § 3 for definitions and guarantees).

**Positioning Relative to Prior Paradigms.** Standard pipelines typically adjudicate edges either via hypothesis tests for CMI with user-chosen significance levels or by optimizing in-sample objectives with parametric penalties. PEP distinguishes itself along three axes: (i) *Evidence estimation strategy:* Instead of relying on in-sample metrics or marginal tests, PEP employs a prequential, context-aware edge score. This score targets the oracle CMI and achieves statistical stability through sample splitting (cross-fitting). (ii) *Decision criterion:* We replace heuristic thresholds with a computable MDL penalty. This gate explicitly accounts for the combinatorial complexity of the search space restricted by the topological order, rather than merely penalizing the parametric dimension. (iii) *Applicability:* PEP is designed to be compatible with amortized or nonparametric predictors without requiring global likelihood optimization. A broader discussion of related paradigms, including continuous optimization and Bayesian structure learning, is provided in Appendix B.5.

## 3 THE PREQUENTIAL EVIDENCE PRUNING (PEP) FRAMEWORK

We consider independent and identically distributed (i.i.d.) observations $X = (X_1, \ldots, X_d) \sim p$ generated by a Structural Causal Model (SCM) compatible with an unknown Directed Acyclic Graph (DAG) $G^\star$. We explicitly denote $\mathbb{E}[\cdot]$ as the expectation with respect to the true data-generating distribution $p$ unless stated otherwise. Given a topological order $\pi$, the pruning problem reduces to selecting, for each node $j$, the subset of parents from the candidate set $\text{Pred}_\pi(j)$. PEP resolves this decision locally by combining a prequential and context-aware evidence statistic with a computed Minimum Description Length (MDL) gate. This approach maintains the computational efficiency of ordering-based search while providing robust edge selection.

**Prequential Scoring via Sample Splitting.** To ensure statistical validity, we employ a sample-splitting strategy. We partition the sample indices $\{1, \ldots, n\}$ into $K$ disjoint folds $\{I_k\}_{k=1}^K$. For any held-out index $s \in I_k$, the predictive density $\log q_{j,S}(x_j^{(s)} \mid x_S^{(s)})$ is evaluated using a predictor trained exclusively on the complementary set $I_k^c$. This out-of-sample evaluation strategy serves two purposes: it mitigates in-sample optimism and, conditional on the fitted predictors, ensures that the per-sample contributions are statistically independent across $s$. This independence property is essential for the finite-sample concentration guarantees presented in § 3.2.

### 3.1 DEFINITION: THE PREQUENTIAL LOG-EVIDENCE GAIN

For a candidate edge $i \to j$ evaluated within a context $S \subseteq \text{Pred}_\pi(j)$ (where $i \in S$), we define the per-sample prequential log-evidence gain as:

$$\delta_{i \to j}(q; S) = \frac{1}{n} \sum_{s=1}^n \left\{ \log q_{j,S}\big(x_j^{(s)} \mid x_S^{(s)}\big) - \log q_{j,S \setminus \{i\}}\big(x_j^{(s)} \mid x_{S \setminus \{i\}}^{(s)}\big) \right\}. \tag{1}$$

The statistic $\delta_{i \to j}$ quantifies the improvement in predictive log-likelihood, measured in nats per sample, resulting from the inclusion of $X_i$ in the parent set of $X_j$ given the co-parents $S \setminus \{i\}$. Unlike marginal tests, this conditional formulation enables the detection of non-additive interactions and synergies that emerge only when specific variables are observed jointly.

### 3.2 THEORETICAL GUARANTEES

We establish the theoretical properties of PEP under the following standing assumptions.

**Assumption 1** (Data and regularity)**.** *(i) The samples $x^{(1)}, \ldots, x^{(n)}$ are independent and identically distributed (i.i.d.) according to $p$. (ii) For all $S \subseteq \text{Pred}_\pi(j)$, both the true conditional density*

$p(x_j \mid x_S)$ *and the predictive density* $q_{j,S}(x_j \mid x_S)$ *have finite log-loss and variance. (iii) All likelihood terms are evaluated using the sample-splitting (prequential) procedure described in* § 3. *Unless stated otherwise, all logarithms are natural and code lengths are measured in nats.*

**Theorem 1** (Population target equals CMI). *With an ideal predictor* $q = p$, *the expected evidence gain satisfies:*

$$\mathbb{E}\big[\delta_{i \to j}(p; S)\big] \;=\; I\big(X_j; X_i \mid X_{S \setminus \{i\}}\big).$$

*Proof sketch.* Taking expectations in Eq. (1) with $q = p$ yields the difference of conditional entropies $-H(X_j \mid X_S) + H(X_j \mid X_{S \setminus \{i\}})$. By the chain rule of mutual information, this equality simplifies to $I(X_j; X_i \mid X_{S \setminus \{i\}})$. Full details are provided in Appendix E.1. □

The statistic maintains stability even with imperfect predictors. Its deviation from the oracle target is bounded by the conditional log-loss regrets of the competing predictive families.

**Proposition 1** (Stability under log-loss regret). *Let* $r_S = \mathbb{E}[\log p(X_j \mid X_S) - \log q_{j,S}(X_j \mid X_S)] \geq 0$ *denote the regret, and define* $r_{S \setminus \{i\}}$ *analogously. Then, the following bound holds:*

$$\big|\mathbb{E}[\delta_{i \to j}(q; S)] - \mathbb{E}[\delta_{i \to j}(p; S)]\big| \;\leq\; r_S + r_{S \setminus \{i\}}.$$

*Proof sketch.* We add and subtract the oracle terms and rearrange the expression. See Appendix E.2 for a formulation based on Bregman divergence. □

To control finite-sample fluctuations, we define the per-sample log-likelihood differences as $Z_s = \log q_{j,S}(X_j^{(s)} \mid X_S^{(s)}) - \log q_{j,S \setminus \{i\}}(X_j^{(s)} \mid X_{S \setminus \{i\}}^{(s)})$ and assume they exhibit sub-exponential tails uniformly in $s$.

**Theorem 2** (Concentration under prequential scoring). *Assume that the random variables* $\{Z_s\}$ *are sub-exponential with parameters* $(\nu, b)$ *and are computed using the prequential procedure. Then, for any* $t > 0$,

$$\Pr\Big(\big|\delta_{i \to j}(q; S) - \mathbb{E}[\delta_{i \to j}(q; S)]\big| \geq t\Big) \;\leq\; 2\exp\Big(-c\,n\,\min\{t^2/\nu^2,\; t/b\}\Big),$$

*where* $c > 0$ *is an absolute constant.*

*Proof sketch.* Conditional on the predictors fitted on complementary folds, the terms $\{Z_s\}$ become independent across $s$. We apply Bernstein's inequality to these independent terms and then remove the conditioning using the tower property. See Appendix E.3 for detailed derivations and an extension to uniform bounds over the edge set. □

Furthermore, if the sub-exponential parameters hold uniformly over forward candidates, a union bound yields a uniform tail bound over the edge set (see Appendix E.3). This result has two immediate practical implications. First, in the absence of a contextual signal, the statistic concentrates near zero.

**Corollary 1** (Null behavior). *If* $X_j \perp X_i \mid X_{S \setminus \{i\}}$ *and the regrets are small, then* $\delta_{i \to j}(q; S)$ *concentrates near* 0 *at the rate specified in* Thm. 2.

*Proof sketch.* This follows by combining Thm. 1 (which states the oracle target is 0 under conditional independence), Proposition 1 (the bias bound), and Thm. 2. □

Second, the decision rule provides finite-sample control when the expected evidence separates true and false edges by a margin.

**Corollary 2** (Finite-sample decision under a margin). *Fix a node* $j$ *and context sets* $\{S_{ij}\}$ *for candidates* $i \in \mathrm{Pred}_\pi(j)$, *where* $P_j = |\mathrm{Pred}_\pi(j)|$ *denotes the number of admissible predecessors of node* $j$ *in the topological order. Suppose there exists a margin* $\gamma > 0$ *such that* $\mathbb{E}[\delta_{i \to j}(q; S_{ij})] \geq \tau_j^{\mathrm{MDL}}(S_{ij}, i) + \gamma$ *for all true parents, and* $\mathbb{E}[\delta_{i \to j}(q; S_{ij})] \leq \tau_j^{\mathrm{MDL}}(S_{ij}, i) - \gamma$ *for all non-parents. If the sub-exponential condition holds uniformly with parameters* $(\nu, b)$, *then the probability of making any inclusion or exclusion error at node* $j$ *is at most* $2P_j \exp(-c\,n\,\min\{\gamma^2/\nu^2, \gamma/b\})$.

*Proof sketch.* We apply Thm. 2 to each candidate edge and apply a union bound over the $P_j$ candidates. See Appendix E.4 for details. □

---

**Algorithm 1** Prequential Evidence Pruning (given topological order $\pi$). The hierarchical group variant described in Section 3.4 utilizes the same decision rule but applies it to groups of parents.

---

1: **Input:** dataset $D$, topological order $\pi$, predictive component $q$, fold indices $\{I_k\}_{k=1}^K$.
2: **Initialize:** For each node $j$, set $S_j \leftarrow \text{Pred}_\pi(j)$.
3: **for** each node $j$ in topological order $\pi$ **do**
4:     **for** each candidate $i \in S_j$ (sorted by marginal affinity) **do**
5:         Compute the prequential log-likelihoods and the resulting gain $\delta_{i\rightarrow j}$ using Eq. (1).
6:         Compute the threshold $\tau \leftarrow \tau_j^{\text{MDL}}(S_j, i)$ using Eq. (3) with structural penalty $\Omega(n, d)$.
7:         **if** $\delta_{i\rightarrow j} \leq \tau$ **then**
8:             Prune edge $(i, j)$ and update $S_j \leftarrow S_j \setminus \{i\}$.
9:         **end if**
10:     **end for**
11: **end for**
12: **Output:** pruned DAG $\widehat{G}$.

---

## 3.3 THE ADAPTIVE STRUCTURAL MDL GATE

To convert the prequential evidence gain into a robust binary pruning decision, we require a principled threshold that balances predictive improvement against model complexity. Since fixed thresholds fail to generalize across varying sample sizes $n$ and graph dimensions $d$, we introduce the *Adaptive Structural MDL Gate*, grounded in the principles of the Extended Bayesian Information Criterion (EBIC).

An edge is retained if and only if the data-compression gain exceeds the adaptive code-length cost required to describe the structural change:

$$\text{Keep edge } i \rightarrow j \quad \Longleftrightarrow \quad \delta_{i\rightarrow j}(q; S) > \tau_j^{\text{MDL}}(S, i). \tag{2}$$

The adaptive threshold $\tau_j^{\text{MDL}}$ is formulated as:

$$\tau_j^{\text{MDL}}(S, i) = \frac{1}{n}\left\{\underbrace{\ln(P_j - k)}_{\text{Identity Cost}} + \underbrace{\ln(k + 1)}_{\text{Sparsity Cost}} + \underbrace{\Omega(\mathbf{n}, \mathbf{d})}_{\text{Structural Penalty}}\right\}, \tag{3}$$

where $k = |S \setminus \{i\}|$ denotes the current parent set size and $P_j = |\text{Pred}_\pi(j)|$ represents the number of admissible candidates. The first two terms encode the costs for identifying the specific parent and specifying the new set size, respectively. See details in Appendix B.1–Appendix B.2

The core innovation lies in the structural complexity penalty $\Omega(\mathbf{n}, \mathbf{d})$, which allows the framework to scale robustly. We formulate this penalty as a multiplicative interaction:

$$\Omega(\mathbf{n}, \mathbf{d}) = \underbrace{\eta}_{\text{Strength}} \cdot \underbrace{\ln n}_{\text{Confidence}} \cdot \underbrace{\ln d^2}_{\text{Complexity}}. \tag{4}$$

This formulation bundles three complementary safeguards. The model confidence term $\ln n$ ensures asymptotic consistency by preventing spurious correlations from crossing the decision boundary as the sample size grows. The search space complexity term $\ln d^2$ acts as an Extended BIC correction for the quadratic number of candidate edges $|\mathcal{E}_\pi| \leq d^2/2$, effectively raising the evidence barrier in high-dimensional regimes to control the family-wise error rate. Finally, the scaling factor $\eta$ calibrates the overall penalty magnitude to trade off precision and recall according to the domain's noise regime.

## 3.4 SCALABLE INFERENCE VIA HIERARCHICAL GROUP PRUNING

To extend PEP to large-scale causal discovery, we introduce a *Hierarchical Group Pruning* strategy. While the sequential backward elimination in Algorithm 1 effectively detects synergies, its quadratic scaling poses a bottleneck in high-dimensional regimes.

We adopt a divide-and-conquer approach inspired by group testing to assess the *collective* predictive evidence of candidate sets. Candidates are first ranked by a lightweight marginal score (e.g., correlation) to concentrate signals, then recursively bisected. We evaluate the joint prequential evidence

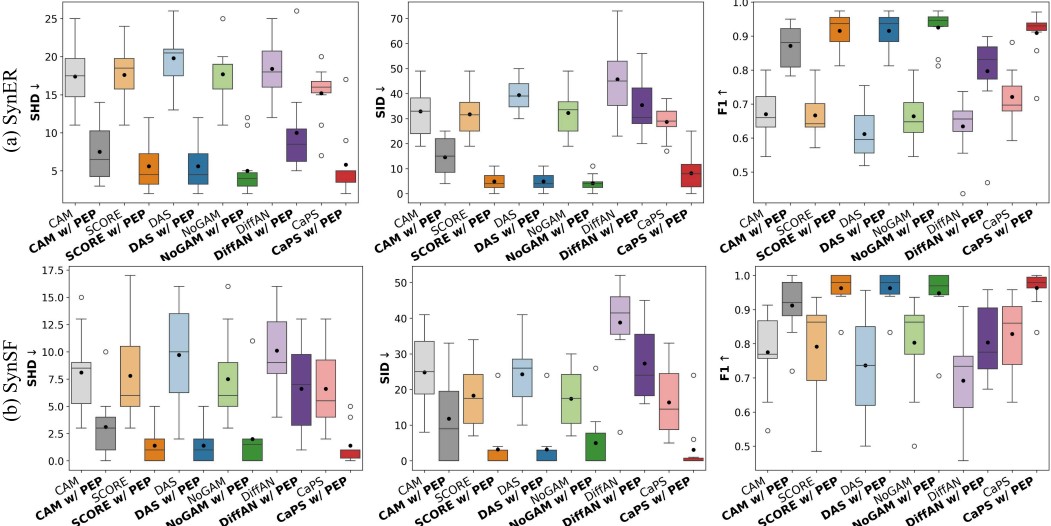

Figure 2: Quantitative comparison of structure learning performance across six ordering-based backbones. The plots contrast the baseline pipelines (utilizing their default marginal pruning) against the PEP-augmented versions on Erdős-Rényi (SynER) and Scale-Free (SynSF) graphs. Lower values are better for SHD and SID; higher values are better for F1.

of each group against the adaptive MDL gate derived from Eq. (3) (details in Appendix B.4). If a group's evidence falls below the threshold, the entire block is pruned simultaneously. Conversely, groups exceeding the threshold are split and re-tested at finer granularity until reduced to individual parents, at which point the standard PEP rule applies.

This strategy significantly reduces pruning complexity. In a sparse regime where a node has at most $s$ true parents ($s \ll P_j$), the number of evidence evaluations scales as $\mathcal{O}(s \log P_j)$ rather than $\mathcal{O}(P_j^2)$. Intuitively, only groups containing true parents trigger subdivisions, creating a logarithmic-depth search tree. For example, with $P_j = 50$ candidates and sparse connectivity ($s \approx 3$), hierarchical pruning reduces evaluations from $\approx 1,275$ to just $\approx 17$. This yields substantial speedups while preserving the detection of synergistic interactions, as the MDL decision rule remains consistent across resolutions.

## 4 EXPERIMENTS

**Experimental Setup.** We evaluate PEP as a plug-in module for six ordering backbones across synthetic (Erdős & Rényi, 1960; Bollobás et al., 2003), misspecified (Montagna et al., 2023a), and real-world (Sachs et al., 2005; Van den Bulcke et al., 2006) benchmarks. To ensure reliability, all results are averaged over 10 independent runs using standard metrics: Structural Hamming Distance (SHD), Structural Intervention Distance (SID), and F1-score. A comprehensive description of the experimental protocol, including dataset generation details and backbone configurations, is provided in Appendix F.

**Plug-and-Play Improvements Across Ordering Backbones.** While research on ordering-based causal discovery has seen significant advancements in topological sort estimation, the subsequent pruning stage has remained largely static, predominantly relying on the standard CAM-pruning procedure. We hypothesized that this reliance on marginal testing constitutes an overlooked bottleneck that constrains the potential of even the most sophisticated ordering algorithms. To demonstrate that PEP resolves this limitation, we replaced the default pruning modules of six state-of-the-art backbones with our framework. The results in Fig. 2 show a clear and consistent pattern: regardless of the underlying ordering algorithm or graph topology (ER or SF), the PEP-augmented pipelines systematically outperform their original counterparts. This substantial reduction in SHD and SID confirms that upgrading the pruning stage from marginal to context-aware evidence evaluation is essential to fully realize the capabilities of modern ordering-based methods.

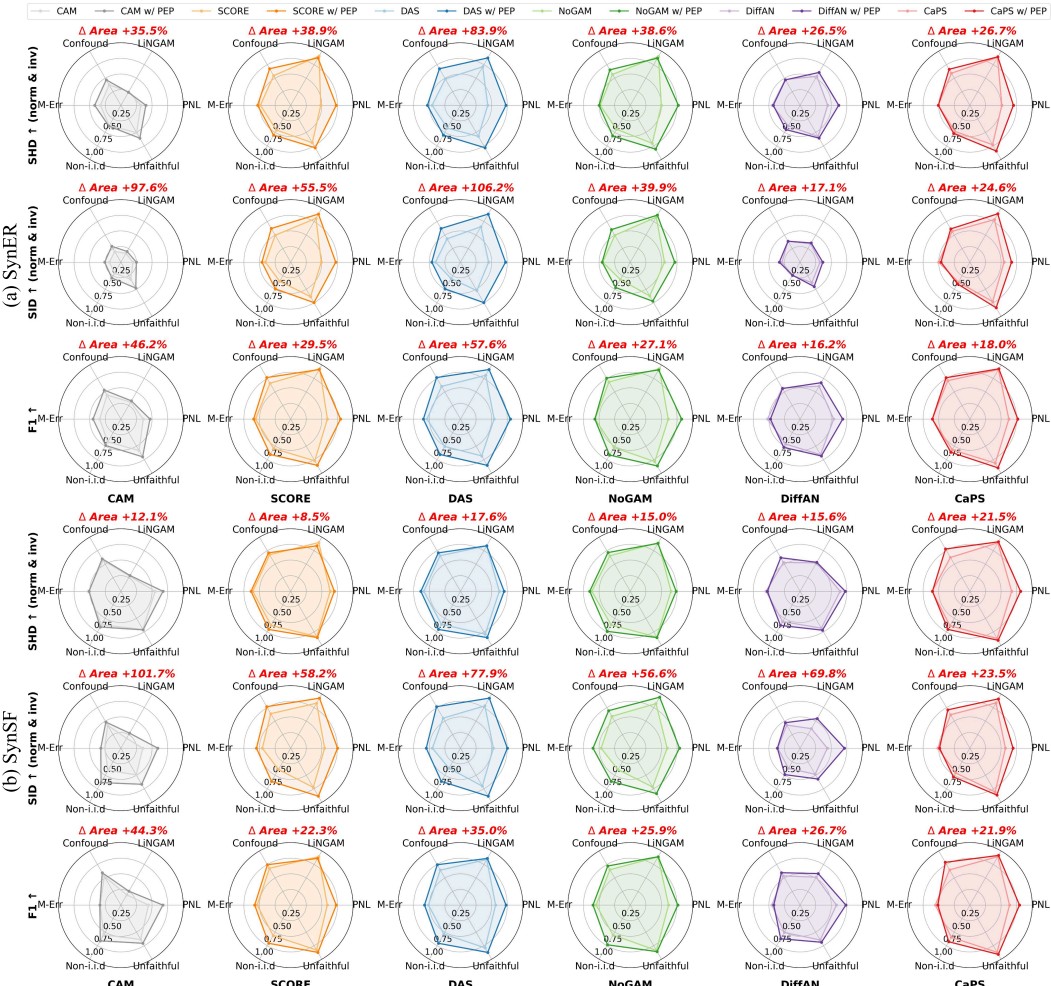

Figure 3: Comparison under model misspecification scenarios. Each radar chart visualizes structural accuracy using three normalized axes: inverted SHD & SID, and F1 score (larger areas indicate better performance). The legend reports the relative area growth rate of PEP compared to the baseline CAM pruning, quantifying the robustness gain across six distinct data-generating mechanisms.

**Robustness Under Misspecification.** Standard causal discovery algorithms often rely on strict assumptions such as additivity or causal sufficiency, which rarely hold in complex real-world systems. We hypothesized that PEP's information-theoretic criterion would remain robust even when these structural assumptions are violated. To test this, we conducted a stress test across six scenarios shown in Fig. 3. The empirical results reveal a decisive advantage for PEP, which is most pronounced in the Post-Nonlinear (PNL) setting. In this regime, the data-generating process explicitly breaks the additivity assumption required by standard marginal pruning. While baseline methods degrade significantly due to their reliance on rigid functional forms, PEP successfully recovers these complex dependencies. This confirms that our context-aware evaluation, which approximates Conditional Mutual Information via a flexible density estimator, effectively transcends the constraints of traditional approaches. Furthermore, the consistent superiority of PEP under measurement error and confounding demonstrates the versatility of replacing brittle statistical tests with a general MDL principle that adapts to the underlying data distribution.

**Performance on Real-World Benchmarks.** To assess practical utility beyond synthetic data, we evaluated PEP on the Sachs protein-signaling network and the SynTReN gene expression benchmark using the CaPS backbone (results in Table 1). On the Sachs dataset, PEP maintains parity with state-of-the-art performance, confirming that our principled approach incurs no degradation on established

Table 1: Quantitative comparison on real-world datasets (Sachs and SynTReN). Best results are high-lighted in bold. (Standard deviations in parentheses).

| Dataset | Sachs | | | SynTReN | | |
|---|---|---|---|---|---|---|
| Metrics | SHD ↓ | SID ↓ | F1 ↑ | SHD ↓ | SID ↓ | F1 ↑ |
| CAM | $12.0_{(0.0)}$ | $55.0_{(0.0)}$ | $0.44_{(0.00)}$ | $41.3_{(9.9)}$ | $170.2_{(45.2)}$ | $0.22_{(0.09)}$ |
| SCORE | $12.0_{(0.0)}$ | $45.0_{(0.0)}$ | $0.44_{(0.00)}$ | $38.6_{(7.0)}$ | $187.5_{(58.6)}$ | $0.21_{(0.09)}$ |
| DAS | $13.0_{(0.0)}$ | $48.0_{(0.0)}$ | $0.33_{(0.00)}$ | $39.4_{(8.0)}$ | $168.3_{(55.4)}$ | $0.23_{(0.07)}$ |
| NoGAM | $12.0_{(0.0)}$ | $45.0_{(0.0)}$ | $0.44_{(0.00)}$ | $39.2_{(7.0)}$ | $184.9_{(59.9)}$ | $0.20_{(0.08)}$ |
| DiffAN | $13.0_{(1.6)}$ | $50.3_{(7.6)}$ | $0.36_{(0.15)}$ | $41.4_{(6.9)}$ | $196.7_{(74.7)}$ | $0.19_{(0.11)}$ |
| CaPS | $11.0_{(0.0)}$ | $42.0_{(0.0)}$ | $0.50_{(0.00)}$ | $37.2_{(5.3)}$ | $178.9_{(58.6)}$ | $0.23_{(0.07)}$ |
| PEP | $11.0_{(0.0)}$ | $42.0_{(0.0)}$ | $\mathbf{0.50}_{(0.00)}$ | $\mathbf{33.0}_{(7.7)}$ | $\mathbf{164.3}_{(26.6)}$ | $\mathbf{0.24}_{(0.03)}$ |

Table 2: Impact of pruning strategy under a non-informative random topological order. This setting isolates the pruning performance from the ordering quality.

| Dataset | Method | SHD ↓ | SID ↓ | F1 ↑ |
|---|---|---|---|---|
| SynER | CAM pruning | $26.0_{(4.6)}$ | $72.4_{(5.4)}$ | $0.39_{(0.11)}$ |
| | PEP | $\mathbf{24.6}_{(7.4)}$ | $\mathbf{68.0}_{(9.3)}$ | $\mathbf{0.44}_{(0.18)}$ |
| SynSF | CAM pruning | $19.0_{(6.4)}$ | $59.4_{(15.3)}$ | $0.50_{(0.15)}$ |
| | PEP | $\mathbf{17.6}_{(7.4)}$ | $\mathbf{58.8}_{(15.7)}$ | $\mathbf{0.50}_{(0.18)}$ |

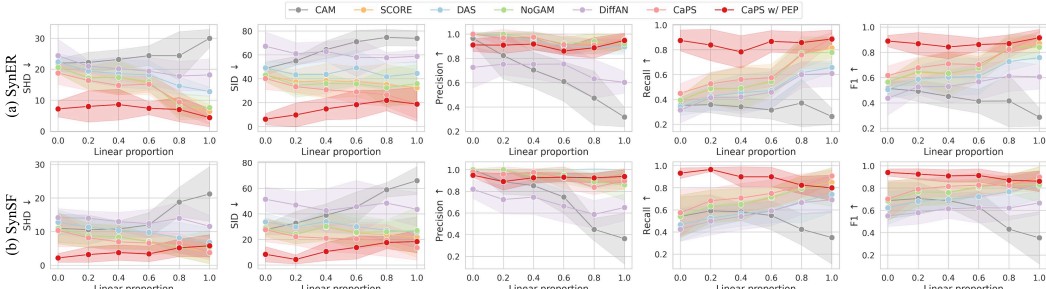

Figure 4: Impact of functional form. We evaluate robustness by sweeping the linearity probability $\rho_{\mathrm{lin}} \in [0, 1]$. For each node, the causal mechanism is generated as a linear function with probability $\rho_{\mathrm{lin}}$ and as a non-linear function with probability $1 - \rho_{\mathrm{lin}}$.

tasks. More importantly, on the challenging SynTReN dataset, PEP delivers a statistically significant improvement in structural accuracy (SHD). These results indicate that PEP is a robust module: it preserves reliability on standard benchmarks while offering decisive advantages in complex, noisy real-world scenarios.

**Isolating Pruning Performance via Random Ordering.** We sought to decouple the efficacy of the pruning stage from the quality of the topological ordering. To this end, we evaluated performance using a random topological order, a worst-case scenario where the pruner must identify the true structure from a dense supergraph of all possible forward edges without informative ordering cues. As shown in Table 2, PEP consistently outperforms standard CAM pruning on both ER and SF graphs. This experiment confirms that the performance gains of PEP are not merely inherited from a strong ordering backbone but are intrinsic to its local evidence-versus-complexity decision rule.

**Robustness to Functional Form Mechanisms.** A core advantage of PEP is its theoretical independence from specific functional forms, unlike marginal tests that often assume linearity. To verify this adaptability empirically, we varied the linearity probability $\rho_{\mathrm{lin}} \in [0, 1]$ in the data-generating process. Specifically, each structural assignment $X_j := f_j(\mathrm{Pa}_j) + \epsilon_j$ is chosen to be a linear function with probability $\rho_{\mathrm{lin}}$ and a non-linear function with probability $1 - \rho_{\mathrm{lin}}$. As shown in Fig. 4, PEP delivers decisive gains in complex, mixed-linearity regimes ($\rho_{\mathrm{lin}} \approx 0.5$) where traditional methods falter. Crucially, even in predominantly linear settings ($\rho_{\mathrm{lin}} \to 1.0$) where CAM pruning is theoretically optimal, PEP remains highly competitive. This demonstrates that our framework incurs no performance penalty when the problem simplifies, effectively bridging the gap between complex and simple causal mechanisms.

**Robustness to the Predictive Component.** To disentangle the algorithmic contribution of our framework from the inductive bias of the density estimator, we instantiated PEP with a diverse suite of predictors: Random Forest (Breiman, 2001), XGBoost (Chen & Guestrin, 2016), Cat-Boost (Prokhorenkova et al., 2018), LightGBM (Ke et al., 2017), and MITRA (Zhang et al., 2025). For these standard estimators, we applied Platt scaling to ensure they provide calibrated probabilistic outputs. As shown in Table 3, PEP consistently improves performance over the CAM-pruning baseline across this broad spectrum of estimators. This validates a core theoretical premise: the effectiveness

Table 3: Performance comparison of PEP instantiated with various predictive components on the SynER dataset ($d = 10$). We report mean and standard deviation (subscript). **Bold** indicates the best performance, and underline indicates the second best. The Avg. Rank is calculated across all 15 row scenarios (5 orderings × 3 metrics).

| Metric | Ordering | CAM-pruning (Base) | PEP w/ Various Predictors | | | | | |
|---|---|---|---|---|---|---|---|---|
| | | | RF | XGBoost | CatBoost | LightGBM | MITRA | TabPFN |
| SHD↓ | CAM | $17.1_{(3.6)}$ | $\underline{10.7}_{(3.7)}$ | $15.5_{(5.2)}$ | $11.7_{(3.6)}$ | $13.0_{(2.9)}$ | $13.3_{(3.4)}$ | $\mathbf{9.9}_{(3.5)}$ |
| | SCORE | $14.9_{(4.1)}$ | $\underline{7.4}_{(2.4)}$ | $12.2_{(2.8)}$ | $7.5_{(2.3)}$ | $7.7_{(1.9)}$ | $10.0_{(3.7)}$ | $\mathbf{5.4}_{(3.4)}$ |
| | NoGAM | $14.9_{(4.0)}$ | $\underline{7.0}_{(2.3)}$ | $11.0_{(2.7)}$ | $6.9_{(2.2)}$ | $7.7_{(2.5)}$ | $9.3_{(3.4)}$ | $\mathbf{4.7}_{(3.2)}$ |
| | DiffAN | $16.0_{(4.0)}$ | $\underline{9.4}_{(3.4)}$ | $13.6_{(3.9)}$ | $9.9_{(3.1)}$ | $10.7_{(2.5)}$ | $10.4_{(3.4)}$ | $\mathbf{7.3}_{(3.1)}$ |
| | CaPS | $15.2_{(3.9)}$ | $\underline{8.9}_{(2.9)}$ | $12.3_{(2.5)}$ | $8.6_{(2.1)}$ | $9.3_{(3.0)}$ | $11.2_{(3.0)}$ | $\mathbf{6.8}_{(3.3)}$ |
| SID↓ | CAM | $42.6_{(7.3)}$ | $33.9_{(14.9)}$ | $33.0_{(12.6)}$ | $\mathbf{30.5}_{(9.5)}$ | $33.7_{(10.0)}$ | $35.1_{(9.8)}$ | $\underline{30.8}_{(8.8)}$ |
| | SCORE | $26.6_{(8.2)}$ | $8.2_{(4.2)}$ | $11.4_{(5.2)}$ | $6.6_{(4.0)}$ | $\mathbf{6.4}_{(2.8)}$ | $9.9_{(4.6)}$ | $\underline{8.5}_{(5.9)}$ |
| | NoGAM | $26.6_{(8.0)}$ | $8.3_{(4.2)}$ | $10.3_{(4.4)}$ | $\underline{5.3}_{(3.3)}$ | $\mathbf{4.5}_{(2.4)}$ | $7.9_{(4.1)}$ | $6.0_{(4.8)}$ |
| | DiffAN | $36.5_{(11.4)}$ | $\underline{16.5}_{(12.8)}$ | $20.6_{(11.7)}$ | $15.4_{(11.5)}$ | $17.2_{(11.6)}$ | $18.8_{(11.8)}$ | $20.5_{(12.3)}$ |
| | CaPS | $28.4_{(7.9)}$ | $12.1_{(6.0)}$ | $13.6_{(4.8)}$ | $10.8_{(4.8)}$ | $\mathbf{9.7}_{(5.7)}$ | $\underline{13.5}_{(4.5)}$ | $14.6_{(8.4)}$ |
| F1↑ | CAM | $0.67_{(0.07)}$ | $\underline{0.80}_{(0.08)}$ | $0.73_{(0.11)}$ | $0.79_{(0.07)}$ | $0.77_{(0.06)}$ | $0.75_{(0.08)}$ | $\mathbf{0.81}_{(0.07)}$ |
| | SCORE | $0.73_{(0.08)}$ | $\underline{0.89}_{(0.04)}$ | $0.82_{(0.05)}$ | $\underline{0.89}_{(0.03)}$ | $\underline{0.89}_{(0.03)}$ | $0.85_{(0.06)}$ | $\mathbf{0.91}_{(0.06)}$ |
| | NoGAM | $0.73_{(0.07)}$ | $\underline{0.90}_{(0.03)}$ | $0.84_{(0.04)}$ | $\underline{0.90}_{(0.03)}$ | $0.89_{(0.04)}$ | $0.87_{(0.05)}$ | $\mathbf{0.93}_{(0.05)}$ |
| | DiffAN | $0.70_{(0.07)}$ | $0.85_{(0.06)}$ | $0.79_{(0.07)}$ | $0.85_{(0.06)}$ | $0.83_{(0.05)}$ | $0.84_{(0.06)}$ | $\mathbf{0.87}_{(0.06)}$ |
| | CaPS | $0.72_{(0.07)}$ | $0.86_{(0.05)}$ | $0.82_{(0.04)}$ | $0.87_{(0.03)}$ | $0.86_{(0.05)}$ | $0.83_{(0.06)}$ | $\mathbf{0.88}_{(0.06)}$ |
| **Avg. Rank** | | 7.00 | 2.47 | 5.80 | 2.53 | 3.07 | 4.73 | **2.40** |

Table 4: Scalability analysis on synthetic datasets with increasing graph sizes ($d \in \{30, 50, 100\}$) and an expected edge count of $4d$. We compare standard pruning methods against our proposed PEP framework across various ordering backbones. Results are reported as $\text{Mean}_{(\text{Std})}$. **Bold** numbers denote improved performance (lower SHD/SID, higher F1) achieved by applying PEP.

| Ordering | Pruning | $d = 30$ | | | $d = 50$ | | | $d = 100$ | | |
|---|---|---|---|---|---|---|---|---|---|---|
| | | SHD↓ | SID↓ | F1↑ | SHD↓ | SID↓ | F1↑ | SHD↓ | SID↓ | F1↑ |
| CAM | Base | $74.2_{(12.1)}$ | $499.9_{(83.7)}$ | $0.54_{(0.06)}$ | $139.4_{(18.7)}$ | $1463.2_{(194.5)}$ | $0.48_{(0.06)}$ | $275.5_{(24.2)}$ | $6007.9_{(571.5)}$ | $0.47_{(0.04)}$ |
| | PEP | $\mathbf{67.4}_{(13.3)}$ | $\mathbf{391.1}_{(98.7)}$ | $\mathbf{0.62}_{(0.09)}$ | $\mathbf{130.7}_{(24.4)}$ | $\mathbf{1193.4}_{(285.5)}$ | $\mathbf{0.55}_{(0.09)}$ | $\mathbf{267.2}_{(28.7)}$ | $\mathbf{5205.3}_{(269.1)}$ | $\mathbf{0.51}_{(0.04)}$ |
| SCORE | Base | $69.6_{(11.1)}$ | $406.7_{(45.8)}$ | $0.58_{(0.05)}$ | $133.0_{(19.4)}$ | $1287.4_{(127.0)}$ | $0.51_{(0.06)}$ | $263.9_{(26.1)}$ | $5408.4_{(425.3)}$ | $0.50_{(0.04)}$ |
| | PEP | $\mathbf{55.3}_{(12.9)}$ | $\mathbf{269.3}_{(48.6)}$ | $\mathbf{0.70}_{(0.07)}$ | $\mathbf{114.9}_{(21.2)}$ | $\mathbf{974.3}_{(148.5)}$ | $\mathbf{0.62}_{(0.07)}$ | $\mathbf{246.5}_{(23.3)}$ | $\mathbf{4478.4}_{(393.4)}$ | $\mathbf{0.57}_{(0.03)}$ |
| NoGAM | Base | $69.5_{(12.9)}$ | $410.4_{(65.7)}$ | $0.58_{(0.06)}$ | $131.9_{(19.6)}$ | $1249.9_{(169.4)}$ | $0.52_{(0.06)}$ | $264.6_{(25.0)}$ | $5357.6_{(489.5)}$ | $0.50_{(0.04)}$ |
| | PEP | $\mathbf{56.7}_{(12.3)}$ | $\mathbf{247.6}_{(72.0)}$ | $\mathbf{0.71}_{(0.05)}$ | $\mathbf{115.2}_{(23.6)}$ | $\mathbf{947.8}_{(71.9)}$ | $\mathbf{0.61}_{(0.07)}$ | $\mathbf{250.0}_{(26.4)}$ | $\mathbf{4338.4}_{(410.2)}$ | $\mathbf{0.57}_{(0.03)}$ |
| DiffAN | Base | $75.1_{(14.4)}$ | $495.7_{(73.9)}$ | $0.53_{(0.08)}$ | $141.7_{(18.4)}$ | $1560.1_{(168.0)}$ | $0.46_{(0.05)}$ | $284.3_{(26.6)}$ | $6338.7_{(499.8)}$ | $0.45_{(0.04)}$ |
| | PEP | $\mathbf{66.0}_{(16.6)}$ | $\mathbf{405.8}_{(95.7)}$ | $\mathbf{0.62}_{(0.10)}$ | $\mathbf{130.0}_{(17.7)}$ | $\mathbf{1345.8}_{(143.5)}$ | $\mathbf{0.55}_{(0.04)}$ | $\mathbf{273.7}_{(32.4)}$ | $\mathbf{5579.8}_{(620.7)}$ | $\mathbf{0.50}_{(0.05)}$ |
| CaPS | Base | $71.2_{(11.7)}$ | $436.3_{(44.2)}$ | $0.57_{(0.05)}$ | $136.2_{(18.1)}$ | $1348.1_{(129.7)}$ | $0.50_{(0.05)}$ | $276.8_{(29.0)}$ | $5754.1_{(427.7)}$ | $0.47_{(0.04)}$ |
| | PEP | $\mathbf{59.7}_{(13.7)}$ | $\mathbf{327.0}_{(59.6)}$ | $\mathbf{0.66}_{(0.07)}$ | $\mathbf{121.9}_{(24.1)}$ | $\mathbf{1117.7}_{(131.3)}$ | $\mathbf{0.58}_{(0.07)}$ | $\mathbf{265.8}_{(31.8)}$ | $\mathbf{5103.7}_{(479.8)}$ | $\mathbf{0.52}_{(0.05)}$ |

of PEP is driven by the information-theoretic rigor of the context-aware evidence score and the adaptive MDL gate, rather than being an artifact of a single powerful model. While TabPFN achieves the best overall rank due to its superior zero-shot density estimation, the consistent gains across all predictors confirm that our framework is model-class agnostic and robustly distinguishes true causal mechanisms as long as the predictive component yields calibrated uncertainty.

**Scalability to High-Dimensional Graphs.** We validated the scalability of PEP by extending our evaluation to larger synthetic graphs with dimensions increasing from $d = 30$ to $100$. As presented in Table 4, PEP consistently outperforms the baseline pruning across all graph sizes and ordering backbones. Crucially, the performance advantage of PEP over standard pruning becomes more pronounced as the graph dimension grows. For instance, with the SCORE backbone at $d = 100$, PEP reduces SHD by approximately $6.6\%$ and improves F1 by $14\%$. This empirical trend validates the efficacy of our Adaptive Structural MDL Gate in high-dimensional regimes. Since the search space scales quadratically with $d$, the structural penalty $\Omega(n, d) = \eta \ln n \ln d^2$ becomes increasingly pivotal. By dynamically raising the evidence barrier in proportion to the search space complexity, PEP effectively mitigates the risk of false positives that typically plagues fixed-threshold methods in large-scale graphs.

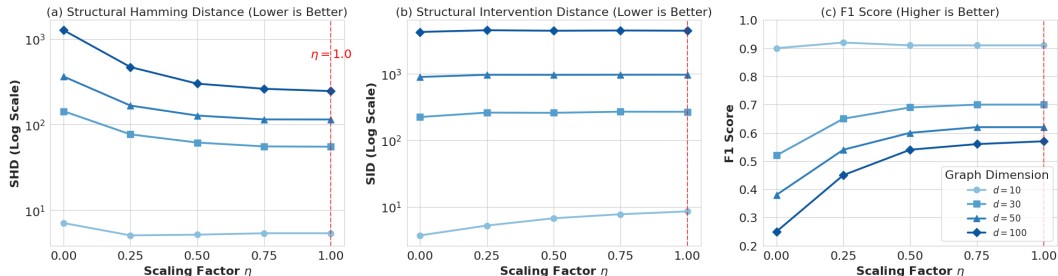

Figure 5: The panels display (a) SHD, (b) SID, and (c) F1 Score across varying graph node dimensions $d \in \{10, 30, 50, 100\}$. The red dashed line marks the theoretical baseline $\eta = 1.0$.

**Empirical Validation of the Structural Penalty.** We examined the sensitivity of the scaling factor $\eta$ to validate the theoretical basis of our structural penalty. As illustrated in Fig. 5, the optimal regularization strength exhibits a critical dependency on the problem scale across all three metrics (SHD, SID, and F1). In low-dimensional settings ($d = 10$), the framework remains robust even with weaker regularization ($\eta < 1.0$). However, in high-dimensional regimes ($d = 100$), performance degrades sharply for small $\eta$, resulting in high SHD and SID values along with a plummeting F1 score. This degradation is driven by an explosion of false positives within the expanded search space when the penalty is insufficient. Crucially, the theoretical baseline of $\eta = 1.0$ consistently achieves optimal performance across all graph sizes and metrics without overfitting. This empirical evidence supports our design choice to fix $\eta = 1.0$ as a robust, parameter-free standard that ensures scalability.

**Computational Efficiency.** We empirically validated the time complexity advantage of our *Hierarchical Group Pruning* by measuring execution times across varying graph dimensions, as shown in Fig. 6. In the low-dimensional regime ($d \leq 30$), the baseline retains a slight edge due to the fixed overhead associated with neural predictor inference and the prequential sample-splitting process. However, as the graph dimension increases, the cubic complexity of the baseline becomes a severe bottleneck. In contrast, PEP demonstrates robust scalability driven by the logarithmic efficiency of group testing. Notably, at $d = 100$, PEP reduces the runtime from $\approx 6,000$ seconds to $\approx 800$ sec-

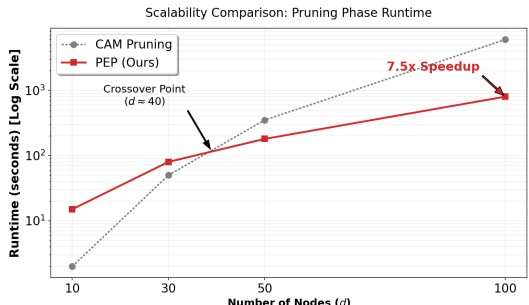

Figure 6: PEP (solid red) demonstrates quasi-linear scaling, surpassing the cubic CAM baseline (dashed grey) at $d \approx 40$.

onds, achieving a $7.5\times$ speedup. This confirms that PEP effectively alleviates the computational bottleneck of existing ordering-based methods, rendering them practically feasible for larger graphs.

# 5 CONCLUSION

In this work, we presented *Prequential Evidence Pruning (PEP)*, a principled framework that fundamentally transforms the pruning stage of causal discovery from heuristic testing to rigorous information-theoretic model selection. By introducing the *Adaptive Structural MDL Gate*, we established a robust, parameter-free decision criterion that dynamically adjusts to varying sample sizes and graph dimensions. This mechanism eliminates the need for manual threshold tuning while effectively controlling false discoveries. Furthermore, our *Hierarchical Group Pruning* successfully resolves the computational bottleneck inherent in traditional backward elimination, reducing the complexity from quadratic to logarithmic and enabling efficient inference on high-dimensional graphs. Extensive empirical validation confirms that PEP achieves state-of-the-art structural accuracy and robustness across diverse ordering backbones and misspecified settings. Consequently, our results demonstrate that PEP not only enhances current ordering-based pipelines but also serves as a scalable and theoretically grounded building block for future advancements in high-dimensional causal discovery.

## REPRODUCIBILITY STATEMENT

We summarize steps taken to ensure reproducibility. Datasets and generation procedures are described in Appendix F.1, the compared backbones and their implementations in Appendix F.2, and evaluation metrics in Appendix F.3. Training and evaluation details, including fold splits and global hyperparameters, are provided in Appendix F. We will release the full codebase and scripts for all experiments upon acceptance to ensure end-to-end reproducibility.

## ETHICS STATEMENT

This work focuses on methodological advances in causal discovery and is evaluated on synthetic benchmarks (SynER and SynSF) and widely used public datasets (Sachs and SynTReN). No personally identifiable information or sensitive attributes are used.

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

# Appendices

# A LLM USAGE

We used large language models only for fixing grammar and typos. All technical content, including theorems, proofs, algorithms, experiments, and analyses, was authored and verified by the paper's authors.

# B ADDITIONAL DETAILS

**Notations.** We summarize symbols used throughout the paper for quick reference. Full definitions are provided in the main text.

Table B.1: Summary of key notations used in the paper.

| Symbol | Definition |
|---|---|
| $\mathcal{E}_\pi, M$ | Forward edge set under order $\pi$, and its size $M = |\mathcal{E}_\pi|$. |
| $X = (X_1, \ldots, X_d), d, n, D$ | Random vector, dimension (#nodes), sample size, and the dataset. |
| $G^\star, G, \widehat{G}, \pi$ | True DAG, a (candidate) graph, pruned DAG, and a topological order. |
| $\mathrm{Pa}_G(j), \mathrm{Ch}_G(j)$ | Parent set and child set of node $j$ in graph $G$. |
| $\mathrm{Pred}_\pi(j), P_j$ | Predecessors of $j$ under order $\pi$; $P_j = |\mathrm{Pred}_\pi(j)|$. |
| $S, S \setminus \{i\} = S', k$ | Working parent set for $j$, the set after removing $i$, and $k = |S'|$. |
| $S_j, m_j$ | Working parent set for node $j$ during pruning; #candidates for $j$ after screening. |
| $p(\cdot), q_{j,S}(\cdot \mid \cdot)$ | True conditional density and predictive conditional density for $X_j \mid X_S$. |
| $q_{j,S}^{(-k)}$ | Out-of-fold predictor for fold $k$ used in prequential scoring. |
| $K, I_k, I_k^c$ | #folds, index set of fold $k$, and its complement. |
| $\delta_{i \to j}(q; S)$ | Prequential log-evidence gain for edge $i \to j$ in context $S$. |
| $r_S$ | Conditional log-loss regret of $q_{j,S}$ relative to $p(\cdot \mid \cdot)$. |
| $Z_s, (\nu, b), c$ | Per-sample log-diff, sub-exponential parameters, and an absolute constant. |
| $\tau_j^{\mathrm{MDL}}(S, i)$ | Adaptive Structural MDL gate. |
| $\Omega(n, d), \eta$ | Structural complexity penalty and its regularization strength scaling factor. |
| $L(\cdot), M_{j,S}$ | Code length in nats; local model for node $j$ with parent set $S$. |
| $I(X; Y \mid Z)$ | Conditional mutual information. |
| $\gamma$ | Margin constant used in finite-sample decision corollaries. |
| $\rho_{\mathrm{lin}}$ | Probability of linear mechanisms in synthetic data generation. |
| $d_S, \Delta d$ | Parametric dimension for context $S$ and its difference . |
| $\Delta\mathrm{BIC}$ | Difference in the Bayesian Information Criterion. |
| $\alpha(r), \bar{\alpha}$ | Per-sample evaluation cost (as a function of parent-set size) and its average. |

## B.1 DERIVATION OF THE TWO-PART CODE FOR LOCAL EDGE ADDITIONS

In ordering-based pruning, we compare the local model for $X_j$ with parent set $S$ against the restricted model with $S' = S \setminus \{i\}$. The description length cost of adding the edge $i \to j$ comprises three transparent information-theoretic components:

1. **Identity Cost** $[\ln(P_j - k)]$: This term encodes the choice of the added parent among the $P_j - k$ remaining admissible candidates from $\mathrm{Pred}_\pi(j)$.

2. **Sparsity Cost** $[\ln(k + 1)]$: Derived from Rissanen's universal code for integers, this term naturally penalizes increasing parent set sizes.

3. **Structural Penalty** $[\mathbf{\Omega(n, d)}]$: This term replaces fixed overhead constants with an adaptive penalty that accounts for the global search space complexity.

Averaging these costs per sample yields the computable adaptive gate:

$$\tau_j^{\mathrm{MDL}}(S, i) = \frac{1}{n}\Big[\ln(P_j - k) + \ln(k + 1) + \mathbf{\Omega(n, d)}\Big],$$

where, as defined in the main text,

$$\mathbf{\Omega(n, d)} = \eta \cdot \ln n \cdot \ln d^2.$$

The structural term $\mathbf{\Omega(n, d)}$ ensures that the decision rule scales consistently with sample size and graph dimension. This approach aligns with Extended BIC-style corrections for multiple comparisons. It recovers local BIC-style comparisons in regular parametric regimes while adapting the complexity penalty to the combinatorial nature of structure learning.

## B.2 FIXED VERSUS ADAPTIVE GATES (SCHEMATIC ILLUSTRATION)

**Adaptive versus fixed gates.** Fig. B.1 visualizes the benefit of the adaptive mechanism. While a fixed threshold might work for a specific dataset scale, it fails as $n$ or $d$ changes. The adaptive MDL gate $\tau_j^{\mathrm{MDL}}(S, i)$ automatically adjusts to the problem complexity: it lowers the per-sample threshold as $n \to \infty$ to recover weak signals (consistency) while raising the structural barrier as $d \to \infty$ to reduce the risk of spurious edges in large graphs. This aligns with our finite-sample concentration result for the prequential statistic (Thm. 2).

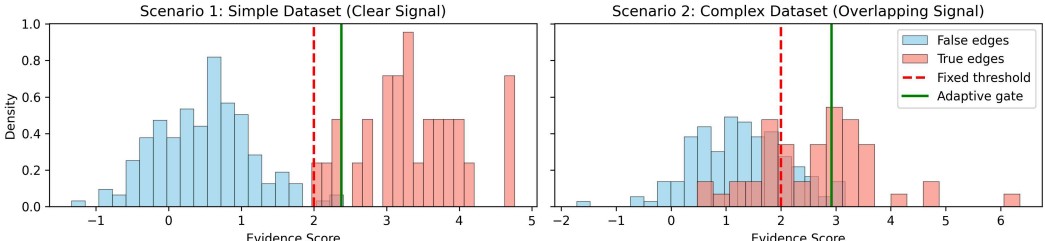

Figure B.1: Fixed versus adaptive gates (schematic). Left: when the distributions of $\delta_{i \to j}$ for true and false edges are well separated, both a fixed threshold and the MDL gate succeed. Right: when the distributions overlap, a fixed threshold erroneously includes many false edges, whereas the MDL gate $\tau_j^{\mathrm{MDL}}(S, i)$ adapts to $(n, P_j, k)$ and maintains separation without validation tuning.

## B.3 RATIONALE AND COMPLEXITY OF BACKWARD ELIMINATION

In this section, we justify the choice of a backward elimination strategy for PEP, analyze its computational complexity, and explain how our Hierarchical Group Pruning strategy mitigates its inherent limitations regarding cost and irrelevant contexts.

**Rationale: The Necessity for Synergy Detection.** A fundamental design choice in PEP is the use of backward elimination (starting with all candidate parents) rather than forward selection. This is driven by the need to detect **non-additive, synergistic interactions** (e.g., the XOR problem or collider structures). In a forward selection approach, candidates are typically evaluated marginally. However, in cases of pure synergy (e.g., $X_j = X_1 \oplus X_2$), the marginal signals are often null ($I(X_j; X_1) \approx 0$). Consequently, a forward search would prematurely discard true parents before their interactive effects could be observed. Backward elimination, by contrast, evaluates each edge $i \to j$ in the context of all other potential parents $S \setminus \{i\}$. This ensures that if $X_1$ and $X_2$ are both present in the context, the conditional evidence $\delta_{1 \to j}(q; S)$ will correctly reflect the strong information gain $I(X_j; X_1 \mid X_2)$, ensuring the retention of synergistic edges.

**Computational Complexity of Standard Backward Elimination.** While theoretically superior for synergy detection, standard greedy backward elimination incurs a high computational cost. For a fixed node $j$, let $P_j = |\mathrm{Pred}_\pi(j)|$ denote the initial number of candidate parents. In the worst-case scenario (e.g., a sparse true graph where most candidates are false positives), the algorithm performs $P_j$ evaluations in the first round, $P_j - 1$ in the second, and so on. The total number of evaluations $N_{eval}$ is:

$$N_{eval} = \sum_{k=1}^{P_j} k = \frac{P_j(P_j + 1)}{2} \in \Theta(P_j^2). \tag{5}$$

Consequently, the total runtime scales quadratically with the number of candidate parents. Specifically for PEP, with $K$-fold cross-fitting and per-sample cost $\bar{\alpha}$, the complexity is $T_{\mathrm{standard}}(j) \in \Theta(K \cdot n \cdot \bar{\alpha} \cdot P_j^2)$. This quadratic scaling becomes a prohibitive bottleneck for high-dimensional graphs ($d \geq 100$), as confirmed in Fig. 6.

**Addressing Limitations via Hierarchical Group Pruning.** Reviewers may rightly concern that starting with a full context containing many irrelevant variables could be computationally expensive and introduce noise. This limitation is precisely what motivates our **Hierarchical Group Pruning** strategy in § 3.4. By recursively testing groups of parents, this strategy addresses both concerns:

1. **Computational Efficiency:** It reduces the complexity from quadratic $\Theta(P_j^2)$ to logarithmic $\mathcal{O}(s \log P_j)$, making the "backward" approach feasible even for large $P_j$.

2. **Noise Reduction:** By pruning entire blocks of irrelevant variables in early group tests, the algorithm rapidly reduces the size of the conditioning set $S$, thereby mitigating the interference from irrelevant variables much faster than examining them one by one.

Thus, PEP leverages backward elimination for its theoretical completeness in capturing synergies, while employing hierarchical pruning to resolve the practical challenges of scalability and noise.

### B.4    COMPLEXITY OF HIERARCHICAL GROUP PRUNING

This section provides a formal complexity analysis of the Hierarchical Group Pruning strategy introduced in § 3.4. For a fixed node $j$, let $P_j = |\operatorname{Pred}_\pi(j)|$ denote the number of candidate parents (predecessors) and let $s$ be the number of true parents (sparsity). Recall from Appendix B.3 that the standard PEP procedure (Algorithm 1) employs sequential backward elimination, which performs $\Theta(P_j^2)$ edge-evaluation tests per node in the worst case. This results in a total computational cost of $\Theta(K n \bar{\alpha} P_j^2)$, where $K$ is the number of folds and $\bar{\alpha}$ is the average per-sample evaluation cost of the predictive component $q$.

The hierarchical variant optimizes this by recursively partitioning the $P_j$ candidates into disjoint groups and applying the MDL decision rule to these sets. We analyze the complexity under the following canonical sparsity assumptions: (i) The initial groups form a balanced binary partition of the $P_j$ candidates. (ii) Any group containing at least one true parent is recursively split until all true parents are isolated. (iii) Any group containing no true parents (null group) is identified and discarded after a constant number of tests, as its prequential evidence falls below the MDL gate.

Under these conditions, the number of PEP evaluations for node $j$ is bounded as follows:

- **Null Group Pruning:** Groups that do not contain any true parents are pruned early. The total number of tests spent on these null groups is proportional to the number of siblings of the active paths, bounded by $\mathcal{O}(s \log P_j)$.
- **Active Search Paths:** Each true parent corresponds to a single path from the root to a leaf in the partition tree. Since the tree height is logarithmic, identifying a single true parent requires $\mathcal{O}(\log P_j)$ group tests.
- **Total Complexity:** With at most $s$ true parents, there are $s$ such paths. Therefore, the total number of group evaluations scales as $\mathcal{O}(s \log P_j)$.
- **Leaf-Level Refinement:** Once a group reduces to a small cluster of individual candidates, the final per-edge pruning incurs only a constant-factor overhead relative to the group search.

Combining these factors, the total number of evidence evaluations $T_j^{\text{group}}$ for node $j$ satisfies:

$$T_j^{\text{group}} \in \mathcal{O}(s \log P_j), \quad \text{assuming } s \ll P_j.$$

This represents a substantial improvement over the quadratic complexity $\Theta(P_j^2)$ of the baseline, particularly in high-dimensional sparse regimes.

Crucially, this hierarchical strategy acts solely as an efficiency enhancement and does not alter the underlying decision logic. Both groups and individual edges are accepted if and only if their prequential gain exceeds the adaptive MDL gate $\tau_j^{\text{MDL}}(S, i)$ defined in Eq. (3). Consequently, the theoretical robustness guarantees established in § 3.2 remain fully applicable to the hierarchical variant.

### B.5    RELATED WORK

**Continuous Optimization & Bayesian Approaches.**    One major paradigm in causal discovery is to cast the problem as a single, continuous optimization problem. This line of work was famously initiated by NOTEARS (Zheng et al., 2018), which introduced a fully differentiable characterization of acyclicity, enabling standard gradient-based methods. This foundational idea was extended by subsequent works to handle non-linear relationships using neural networks, such as GraNDAG (Lachapelle

Table B.2: Comparison of local edge evaluation mechanisms across constraint-based tests, decomposable BIC scoring, and PEP. All code lengths are in nats.

| | Conditional Independence (CI) Tests | Decomposable BIC Scoring | Prequential Evidence Pruning (PEP) |
|---|---|---|---|
| **Core approach** | Decide edges by testing $X_i \perp X_j \mid X_S$ with a user-chosen significance level. | Select a graph by maximizing a global decomposable score that trades off in-sample fit and parametric complexity. | Prune edges under a given order by comparing a local prequential evidence gain with a computed code-length penalty. |
| **Evidence score** | Test statistic $T(X_i, X_j \mid X_S)$ that estimates or surrogates $I(X_j; X_i \mid X_S)$. | Local in-sample log-likelihood difference under decomposability, $\ell(S) - \ell(S \setminus \{i\})$. | Prequential log-evidence gain $\delta_{i \to j}(q; S) = \frac{1}{n} \sum_{s=1}^{n} \log \frac{q_j(x_j^{(s)} \mid x_S^{(s)})}{q_j(x_j^{(s)} \mid x_{S \setminus \{i\}}^{(s)})}$, with $q_j$ evaluated out-of-fold. |
| **Decision rule** | Reject $H_0$ if $p$-value $< \alpha$ (per-test or FDR-controlled). | Accept if $\ell(S) - \ell(S \setminus \{i\}) > \frac{1}{2} \Delta d \frac{\log n}{n}$ (parametric penalty). | Accept if $\delta_{i \to j}(q; S) > \tau_j^{\mathrm{MDL}}(S, i)$, where $\tau_j^{\mathrm{MDL}}(S, i)$ is given by Eq. (3) with $\Omega(n, d) = \eta \ln n \ln d^2$. |
| **Representative properties** | Nonparametric options available; requires $\alpha$; test-by-test decisions and multiple-testing control. | Consistent under correct parametric family; global, in-sample objective; decomposable local updates. | Prequential and context-aware; sample-size aware penalty; no threshold tuning; model-class agnostic. |

et al., 2020) and GOLEM (Ng et al., 2020). Further advancements include DrBO (Duong et al., 2025), which employs sophisticated search strategies like Bayesian optimization, and CGP-CDE (Dhir et al., 2025), which integrates flexible Gaussian Process models. From a more strictly Bayesian perspective, where the goal is to infer a posterior distribution over graphs rather than a single point estimate, methods like DiBS (Lorch et al., 2021) and DECI (Geffner et al., 2024) have been proposed. While powerful, these approaches typically involve complex, model-specific training procedures to learn both the graph and functional parameters.

**Prior-Data Fitted Networks for Causality.** Prior-Data Fitted Networks (PFNs) (Müller et al., 2022) use large-scale, synthetic pre-training to approximate Bayesian predictive inference via in-context learning. TabPFN (Hollmann et al., 2025b) realizes this idea for tabular data and provides calibrated, zero-shot predictive densities that are valuable when samples are scarce or mechanisms are heterogeneous. Building on this paradigm, several works adapt PFNs to *causal inference* tasks. These include models such as FairPFN (Robertson et al., 2025a) for fairness-aware prediction, Do-PFN (Robertson et al., 2025b) for estimating interventional outcomes without a known graph, CausalPFN (Balazadeh et al., 2025) for treatment-effect estimation with calibrated uncertainty, and the comprehensive CausalFM (Ma et al., 2025) framework, illustrating the promise of PFNs as general-purpose causal tools. *In contrast*, we shift the focus to causal discovery. Rather than building an end-to-end PFN for inference, our contribution is a new framework (PEP). It leverages the PFN as a powerful predictive engine to compute a prequential evidence score, which is then assessed by a principled MDL gate.

## C PRELIMINARIES

### C.1 CAUSAL ADDITIVE MODELS (CAM)

CAM (Bühlmann et al., 2014) is a two-stage, ordering-based approach for learning DAGs under an additive structural equation model (SEM). In this framework, each variable is modeled as:

$$X_j = \sum_{k \in \mathrm{pa}(j)} f_{j,k}(X_k) + \varepsilon_j,$$

where the noise terms $\varepsilon_j$ are independent. The learning problem is decomposed into two distinct phases: (i) estimating a topological order and (ii) pruning edges consistent with that order. The key design choice in CAM is to decouple these tasks. The order is estimated by maximizing the restricted likelihood under the additive SEM, whereas sparsity is enforced only during the subsequent pruning step. This separation transforms the intractable structure learning problem into a manageable combination of permutation search and variable selection.

**Stage 1: Order Search.** CAM searches over the space of permutations, optionally restricted by a preliminary skeleton, and selects the order that maximizes the likelihood of the additive SEM. The consistency of this maximum-likelihood order estimator has been established for both low-dimensional and high-dimensional regimes. Intuitively, once the topological order is fixed, the

problem of causal discovery reduces to a set of potentially nonlinear regressions of each node on its predecessors.

**Stage 2: Pruning and Feature Selection.** Given the estimated order, CAM performs variable selection to remove spurious edges. For each node $X_j$, it fits a Generalized Additive Model (GAM) using its predecessors as covariates. It then tests the null hypothesis $H_0 : f_{j,k}(\cdot) \equiv 0$ for each candidate parent $X_k$. Edges that fail to demonstrate a statistically significant contribution at a user-defined level $\alpha$ are discarded.

**Relationship to Conditional Independence Testing.** The pruning mechanism in CAM serves as a marginal, additivity-constrained proxy for a Conditional Independence (CI) test. Conceptually, the null hypothesis $f_{j,k} \equiv 0$ corresponds to the conditional independence statement $X_j \perp\!\!\!\perp X_k \mid \mathrm{Pred}_{\hat{\pi}}(j) \setminus \{k\}$. However, this equivalence holds strictly under the assumption that the true dependencies are additive. Because the test evaluates each parent individually within this additive structure, it acts as a marginal proxy. This creates a critical limitation: CAM pruning cannot capture non-additive synergies, such as XOR-type interactions, where the marginal contribution of a parent may be zero despite a strong joint dependence.

In summary, CAM provides a robust baseline characterized by an efficient likelihood-based order search and a GAM-based pruning step. This precisely delineates the comparison point for our work. While PEP retains the ordering paradigm, we replace the marginal, hypothesis-based pruning of CAM with a joint, context-aware evidence rule designed to overcome the limitations of additivity constraints.

## C.2 Score-based Leaf Identification via the Score Function

Let $s(x) = \nabla_x \log p(x)$ denote the *score function*. Under additive noise models with $X_j = f_j(X_{\mathrm{Pa}(j)}) + \varepsilon_j$ and independent noise terms $\varepsilon_j$, the $j$-th component of the score function decomposes such that the contribution from children nodes vanishes at the leaves. Practical ordering algorithms leverage the properties of the score Jacobian (or the Hessian of the log-likelihood) to iteratively identify and remove leaf nodes:

- **Variance-based (SCORE).** In nonlinear settings, Rolland et al. (2022) demonstrate that a node $X_j$ is a leaf if and only if the variance of the $j$-th diagonal element of the score Jacobian is zero. Based on this, the leaf is identified by minimizing the variance:

$$\hat{j} = \arg\min_j \mathrm{Var}\big[\partial_{x_j} s_j(X)\big].$$

- **Expectation-based (CaPS).** To accommodate both linear and nonlinear relationships robustly, Xu et al. (2024) propose utilizing the expectation of the Jacobian diagonal. A leaf node is identified by maximizing this expected value:

$$\hat{j} = \arg\max_j \mathrm{diag}(\mathbb{E}\left[\nabla s(X)\right]).$$

- **Diffusion-based Estimation (DiffAN).** Sanchez et al. (2023) leverage Denoising Diffusion Probabilistic Models (DDPMs) to scale score estimation, computing the Hessian via backpropagation. To bypass prohibitive retraining after each leaf removal, they introduce the *deciduous score* update. Specifically, the score for the remaining variables is adjusted analytically by subtracting a residue $\Delta_l$:

$$\nabla \log p(x_{-l}) = \nabla \log p(x) - \underbrace{H_{:,l}(\log p(x)) \cdot \frac{\nabla_{x_l} \log p(x)}{H_{l,l}(\log p(x))}}_{\Delta_l}.$$

These criteria yield effective order-estimation subroutines, which we integrate with our proposed pruning module.

## C.3 Prequential Scoring via Sample Splitting

Given a dataset $\{x^{(s)}\}_{s=1}^n$ and a candidate parent set $S \subseteq \mathrm{Pred}_\pi(j)$ for node $j$, let $q_{j,S}$ denote any predictive conditional density estimator for $X_j$ given $X_S$. To ensure statistical independence of the

error terms, we employ a $K$-fold sample-splitting strategy. We partition the indices $\{1, \ldots, n\}$ into disjoint folds $\{I_k\}_{k=1}^K$. For each fold $k$, we fit a predictor on the complementary set $I_k^c$ and evaluate the log-likelihood exclusively on the held-out fold $I_k$:

$$\widehat{\ell}_{\text{preq}}(j, S) = \frac{1}{n} \sum_{k=1}^K \sum_{s \in I_k} \log q_{j,S}^{(-k)}(x_j^{(s)} \mid x_S^{(s)}).$$

This prequential evaluation mitigates in-sample optimism. Furthermore, conditional on the fitted predictors, it renders the per-sample contributions independent across $s$. This independence property is crucial as it enables the application of concentration inequalities for edge-wise evidence differences.

## C.4   CONDITIONAL MUTUAL INFORMATION (CMI)

For random variables $(X, Y, Z)$ with a joint density $p$, the Conditional Mutual Information (CMI) is defined as:

$$I(X; Y \mid Z) = \mathbb{E}\left[\log \frac{p(X \mid Y, Z)}{p(X \mid Z)}\right] = H(X \mid Z) - H(X \mid Y, Z).$$

In the context of PEP, the population target of the prequential log-evidence gain for an edge $i \to j$ equals $I(X_j; X_i \mid X_{S \setminus \{i\}})$ assuming an ideal predictor $q = p$. This theoretical connection justifies the interpretation of our $\delta$ statistic as a context-aware measure of conditional dependence.

## C.5   MINIMUM DESCRIPTION LENGTH (MDL) PRINCIPLE

The Minimum Description Length (MDL) principle formalizes the trade-off between model fit and complexity, effectively quantifying Occam's razor. It posits that the best model is the one providing the shortest lossless description of the data. Using a two-part code, the total length is given by:

$$L(D; M) = L(M) + L(D \mid M),$$

where $L(\cdot)$ denotes the code length in nats. The coding theorem establishes a direct link between code length and probability, specifically $L(x) \approx -\log p(x)$. Consequently, MDL minimizes the sum of the model description cost and the negative log-likelihood of the data. PEP utilizes this principle to derive a decision gate that adapts the penalty based on the combinatorial complexity of the graph structure.

## C.6   STRUCTURAL CAUSAL MODELS (SCMs)

A *Structural Causal Model (SCM)* over $X$ consists of a DAG $G^\star$ and structural assignments

$$X_j = f_j(X_{\text{Pa}_{G^\star}(j)}, \varepsilon_j), \qquad j = 1, \ldots, d,$$

with mutually independent exogenous noises $\varepsilon = (\varepsilon_1, \ldots, \varepsilon_d)$.[1] The induced observational density factorizes as

$$p(x) = \prod_{j=1}^d p(x_j \mid x_{\text{Pa}_{G^\star}(j)}),$$

which is the global Markov property of the DAG. Interventions $do(X_S = x_S)$ replace the assignments $\{f_j : j \in S\}$ by constants and sever incoming edges into $S$, enabling interventional semantics via the truncated factorization. Ordering-based discovery exploits the existence of a (possibly estimated) topological order $\pi$ to constrain candidate parents for $X_j$ to the set $\text{Pred}_\pi(j) = \{i : \pi(i) < \pi(j)\}$ and reduces structure learning to *pruning* spurious edges among these forward links.

## C.7   TABULAR FOUNDATION MODEL (TABPFN) AND PRIOR-DATA FITTED NETWORKS

*Prior-Data Fitted Networks* (PFNs) instantiate in-context learning for supervised tasks by training a transformer to approximate the *Bayesian posterior predictive* over a prior of tasks. A PFN receives, at inference, a full dataset context and emits predictive distributions for held-out points in a single

---

[1]Independence of the exogenous noises (causal sufficiency) may be relaxed to allow latent confounding, but we keep the canonical acyclic, causally sufficient case for clarity.

forward pass. *TabPFN* specializes this idea to tabular data: it is pre-trained on a very large corpus of synthetic datasets sampled from SCM-driven generators spanning mixed data types and diverse mechanisms. Practically, for any $X_j$ and parent set $S$ it returns a calibrated conditional distribution $q_{j,S}(\cdot \mid x_S)$ from which we compute prequential log-likelihoods. For regression with discretized outputs, we integrate the predictive mass over the bin containing the observed value; for categorical data we use the emitted probabilities directly. This zero-shot, calibrated density estimation is what makes TabPFN a convenient predictive component for our framework, eliminating per-dataset training while supporting mixed types.

## D    ILLUSTRATIVE EXAMPLES: WHY CONTEXT-AWARE PRUNING MATTERS

This appendix provides, on concrete mathematical examples, the two claims made in the Introduction and in § 3: (i) pruning must be *context-aware* to capture non-additive structure and to avoid confounding, and (ii) PEP's *computed* MDL gate replaces tuned thresholds with an auditable code-length cost. Each example walks through the marginal calculation (what classical pruning would see) and the PEP calculation (the prequential log-evidence gain $\delta$), then states the decision under the MDL rule $\delta > \tau^{\mathrm{MDL}}$ (Eq. (1)–Eq. (2)). These examples mirror the advantages emphasized in the paper's opening sections and experiments.

**Notations.**    All logarithms are natural (nats). For $p \in (0,1)$, $h(p) = -p \log p - (1-p) \log(1-p)$ denotes the binary entropy. We write $S \subseteq \mathrm{Pred}_\pi(j)$ for the co-parents of $X_j$ (including $i$ when testing $i \to j$). At the oracle ($q = p$), $\mathbb{E}[\delta_{i \to j}(p; S)] = I(X_j; X_i \mid X_{S \setminus \{i\}})$ by Thm. 1; bounded log-loss regret perturbs this by at most $r_S + r_{S \setminus \{i\}}$ (Prop. 1); prequential scoring yields concentration (Thm. 2).

### D.1    NOISY XOR: A CANONICAL CASE OF DISCRETE SYNERGY

We begin with the classic XOR problem, a canonical example where two parents are only informative when considered together. The data is generated by $X_3 = X_1 \oplus X_2 \oplus N$, where the parents $X_1, X_2 \overset{\text{i.i.d.}}{\sim} \mathrm{Bernoulli}(\frac{1}{2})$ and $N \sim \mathrm{Bernoulli}(\varepsilon)$ is a noise term.

A marginal analysis, which evaluates the link $X_1 \to X_3$ in isolation, would find the variables to be independent, as the influence of the random co-parent $X_2$ averages out any effect. This leads to a marginal mutual information of exactly zero:

$$I(X_3; X_1) = 0.$$

A single-parent test would therefore fail. In contrast, PEP's context-aware approach conditions on $X_2$, revealing a clear signal where the oracle evidence gain is strictly positive:

$$\mathbb{E}[\delta_{1 \to 3}(p; \{1,2\})] = I(X_3; X_1 \mid X_2) = \ln 2 - h(\varepsilon) > 0.$$

This demonstrates that while the marginal signal is null, the conditional signal is strong, allowing our proposed method to correctly identify the synergistic relationship.

### D.2    MULTIPLICATIVE INTERACTION: A CASE OF CONTINUOUS SYNERGY

To show this principle extends beyond discrete cases, we consider a continuous synergy defined by $X_3 = X_1 X_2 + \varepsilon$, where parents $X_1, X_2 \overset{\text{i.i.d.}}{\sim} \mathcal{N}(0,1)$ and noise $\varepsilon \sim \mathcal{N}(0, \sigma^2)$. A marginal analysis based on first-order statistics, such as linear regression or covariance, will fail. Because the variables are zero-mean, the marginal covariance is zero:

$$\mathrm{Cov}(X_3, X_1) = 0.$$

A test based on correlation would find no effect. The context-aware approach of PEP, however, targets the CMI by evaluating the full conditional distributions. This is strictly positive and correctly quantifies the information gain from the interaction:

$$\mathbb{E}[\delta_{1 \to 3}(p; \{1,2\})] = I(X_3; X_1 \mid X_2) = \frac{1}{2} \mathbb{E}_{X_2}\left[\log\left(1 + \frac{X_2^2}{\sigma^2}\right)\right] > 0.$$

This confirms that our method can identify purely interactive signals that are invisible to common marginal tests, with an evidence gain that appropriately grows as the noise $\sigma^2$ decreases.

### D.3 Confounding: A Case of Avoiding Spurious Edges

Here we verify that context is crucial for avoiding false positives. Consider a common confounder $C \sim \mathcal{N}(0, 1)$ generating $X_i = aC + \varepsilon_i$ and $X_j = bC + \varepsilon_j$, with no direct edge between them. A marginal analysis will be fooled by the confounder, as the common cause $C$ induces a non-zero spurious correlation:

$$\text{Cov}(X_i, X_j) = ab \, \text{Var}(C) \neq 0.$$

This would lead a marginal method to incorrectly add a non-existent edge. The context-aware approach of PEP avoids this by including the confounder $C$ in the context set. By d-separation, the variables are conditionally independent, and the oracle evidence is exactly zero:

$$\mathbb{E}\big[\delta_{i \to j}(p; \{i, C\})\big] = I(X_j; X_i \mid C) = 0.$$

This verifies that when the confounder is observed, our mechanism correctly finds zero evidence and prunes the spurious edge.

### D.4 Post-Nonlinear Effects: A Case of Robustness to Warping

We next consider a case where a simple relationship is obscured by a non-linear transformation: $X_3 = g(X_1 + X_2 + \varepsilon)$, where $g$ is an invertible, non-linear function. A marginal analysis can be easily fooled. A simple test focused on mean effects might fail because the function $g$ distorts the underlying additive structure. The context-aware approach of PEP is robust to this distortion due to a key property of mutual information: its invariance to invertible transformations. The oracle target for PEP therefore remains strongly positive:

$$I(X_3; X_1 \mid X_2) = I(g(X_1 + X_2 + \varepsilon); X_1 \mid X_2) = I(X_1 + X_2 + \varepsilon; X_1 \mid X_2) > 0.$$

This shows our metric correctly identifies dependencies even when they are obscured by complex transformations.

### D.5 Suppressor Effect: A Case of Handling Collinearity

Finally, we examine the classic suppressor effect, which occurs with highly correlated parents ($\rho \approx 1$) in the model $X_3 = \beta_1 X_1 + \beta_2 X_2 + \varepsilon$, where $\beta_1 \approx -\beta_2$. In a marginal analysis, the effects of the two parents nearly cancel, leading to a marginal covariance close to zero:

$$\text{Cov}(X_3, X_1) = \beta_1 + \beta_2 \rho \approx 0.$$

A marginal test would see a weak signal and might incorrectly prune a true parent. The context-aware approach of PEP resolves this by assessing the contribution of $X_1$ given $X_2$. The conditional signal remains strong, as captured by the CMI:

$$I(X_3; X_1 \mid X_2) = \tfrac{1}{2} \log\Big(1 + \frac{\beta_1^2(1 - \rho^2)}{\sigma^2}\Big) > 0.$$

This demonstrates that our method can identify the true importance of a parent even when its signal is masked by other, highly correlated parents.

### D.6 The Finite-Sample Decision Gate

The preceding examples analyzed the oracle CMI, which represents the ideal signal. This final example connects this theory to the practical, finite-sample decision rule that PEP actually implements. A traditional approach might have a strong evidence metric but still rely on a heuristic or tuned threshold. In contrast, PEP provides an auditable acceptance condition. Our concentration guarantees (Thm. 2) establish a probabilistic lower bound on the empirical evidence $\delta_{i \to j}(q; S)$ that we measure from data. PEP's final step is to keep an edge only if this conservatively estimated signal exceeds the computable MDL penalty, $\tau^{\text{MDL}}$. This transforms the pruning decision into a transparent and principled trade-off, which can be intuitively summarized as:

$$\underbrace{I(X_j; X_i \mid X_{S \setminus \{i\}})}_{\text{Signal}} - \underbrace{(2\varepsilon_{\text{reg}} + \psi_n(\alpha))}_{\text{Uncertainty}} > \underbrace{\tau_j^{\text{MDL}}(S, i)}_{\text{Complexity Cost}} .$$

This provides a complete, theoretically grounded recipe for making a decision, moving beyond the simple identification of a signal.

# E  PROOFS FOR THEORETICAL GUARANTEES

## E.1  POPULATION IDENTITY: PROOF OF THM. 1

*Proof.* Let $S' = S \setminus \{i\}$. Under the ideal predictor assumption $q = p$,

$$\mathbb{E}[\delta_{i \to j}(p; S)] = \mathbb{E}\big[\log p(X_j \mid X_S) - \log p(X_j \mid X_{S'})\big] \tag{6}$$

$$= -H(X_j \mid X_S) + H(X_j \mid X_{S'}) \tag{7}$$

$$= I(X_j; X_i \mid X_{S'}), \tag{8}$$

where the second equality follows from the definition of conditional entropy, and the last equality utilizes the chain rule for conditional mutual information. All expectations are finite by Assumption 1. $\square$

## E.2  STABILITY: PROOF OF PROP. 1

*Proof.* Let $S' = S \setminus \{i\}$. Define $p_S(\cdot) = p(X_j \mid X_S)$ and $q_S(\cdot) = q_{j,S}(X_j \mid X_S)$, and similarly for $S'$. Then,

$$\mathbb{E}[\delta_{i \to j}(q; S)] - \mathbb{E}[\delta_{i \to j}(p; S)] = \mathbb{E}[\log q_S - \log q_{S'}] - \mathbb{E}[\log p_S - \log p_{S'}]$$

$$= \underbrace{\mathbb{E}[\log q_S - \log p_S]}_{-r_S} - \underbrace{\mathbb{E}[\log q_{S'} - \log p_{S'}]}_{-r_{S'}}$$

$$= -r_S + r_{S'}.$$

Consequently, $\big|\mathbb{E}[\delta_{i \to j}(q; S)] - \mathbb{E}[\delta_{i \to j}(p; S)]\big| \le r_S + r_{S'}$. If the regrets satisfy $r_S, r_{S'} \le \varepsilon$, then the bias is bounded by $2\varepsilon$. $\square$

## E.3  CONCENTRATION: PROOF OF THM. 2

*Proof.* Let $Z_s = \log q_{j,S}(X_j^{(s)} \mid X_S^{(s)}) - \log q_{j,S'}(X_j^{(s)} \mid X_{S'}^{(s)})$, where $S' = S \setminus \{i\}$. Consider the $K$-fold sample splitting procedure and denote by $\widehat{q}_{j,S}^{(k)}$ and $\widehat{q}_{j,S'}^{(k)}$ the predictors fitted on the training set $I_k^c$ (complement of fold $k$). Let $\mathcal{F}$ be the $\sigma$-algebra generated by all fitted predictors $\{(\widehat{q}_{j,S}^{(k)}, \widehat{q}_{j,S'}^{(k)})\}_{k=1}^K$. For any index $s \in I_k$, $Z_s$ is a measurable function of the data point $X^{(s)}$ and the predictors $(\widehat{q}_{j,S}^{(k)}, \widehat{q}_{j,S'}^{(k)})$. By construction of the sample splitting, $X^{(s)}$ is independent of the training data used to fit the predictors in $\mathcal{F}$. Therefore, conditional on $\mathcal{F}$, the terms $\{Z_s : s \in [n]\}$ are statistically independent.

Assume that the conditional sub-exponential Orlicz $\psi_1$ norms are uniformly bounded almost surely: $\|Z_s - \mathbb{E}[Z_s \mid \mathcal{F}]\|_{\psi_1} \le c_1 \nu$ and $|Z_s - \mathbb{E}[Z_s \mid \mathcal{F}]| \le c_2 b$ a.s. for constants $(\nu, b)$.[2] Applying the conditional Bernstein's inequality, for any $t > 0$, we have:

$$\Pr\left(\left|\frac{1}{n}\sum_{s=1}^n Z_s - \mathbb{E}[Z_s \mid \mathcal{F}]\right| \ge t \,\middle|\, \mathcal{F}\right) \le 2\exp\left(-cn\min\left\{\frac{t^2}{\nu^2}, \frac{t}{b}\right\}\right).$$

Taking expectations over $\mathcal{F}$ and using the tower property $\mathbb{E}[\mathbb{E}[Z_s \mid \mathcal{F}]] = \mathbb{E}[Z_s]$ yields the unconditional tail bound with the same exponent. Since $\delta_{i \to j}(q; S) = \frac{1}{n}\sum_{s=1}^n Z_s$, the claim follows. $\square$

**Uniform-Over-Edges Extension.** Let $\mathcal{E}_\pi = \{(i, j) : i \in \mathrm{Pred}_\pi(j)\}$ be the set of all candidate forward edges, with $|\mathcal{E}_\pi| = M$. If the sub-exponential parameters $(\nu, b)$ hold uniformly for all edges in $\mathcal{E}_\pi$, then by applying the union bound, we obtain:

$$\Pr\left(\max_{(i,j) \in \mathcal{E}_\pi} \big|\delta_{i \to j}(q; S_{ij}) - \mathbb{E}[\delta_{i \to j}(q; S_{ij})]\big| \ge t\right) \le 2M\exp\left(-cn\min\left\{\frac{t^2}{\nu^2}, \frac{t}{b}\right\}\right),$$

where $S_{ij}$ denotes the co-parent context used for testing the edge $i \to j$.

---

[2] A sufficient condition is that the conditional log-densities are uniformly bounded above, and $q_{j,S}, q_{j,S'}$ are bounded away from 0 on the support of $p$; more generally, it suffices that the conditional Moment Generating Function (MGF) exists in a neighborhood of 0.

### E.4 MDL PENALTY DERIVATION AND FINITE-SAMPLE CONSISTENCY COROLLARY

**Two-Part Code for One-Parent Augmentation.** Let $P_j = |\operatorname{Pred}_\pi(j)|$ and $k = |S \setminus \{i\}|$. Augmenting the parent set from $S' = S \setminus \{i\}$ to $S$ requires encoding two pieces of information: (i) The identity of the added parent among the $P_j - k$ remaining candidates. This can be encoded with a cost of $\ln(P_j - k)$ nats using an optimal prefix code. (ii) The new set size $k + 1$. This contributes a term $\ln(k + 1)$ (up to a constant) under a universal code for integers. We absorb the constant overhead and global structural penalties into the term $\Omega(n, d)$ as defined in Eq. (4) of the main text. Dividing by $n$ yields the local per-sample MDL gate:

$$\tau_j^{\mathrm{MDL}}(S, i) = \frac{1}{n}\Big[ \ln(P_j - k) + \ln(k + 1) + \Omega(n, d) \Big].$$

**Corollary 3** (Finite-Sample Consistency under a Margin). *Fix a node $j$ and context sets $\{S_{ij}\}$ for testing candidates $i \in \operatorname{Pred}_\pi(j)$. Suppose there exists a margin $\gamma > 0$ such that:*

$$\mathbb{E}[\delta_{i \to j}(q; S_{ij})] \geq \tau_j^{\mathrm{MDL}}(S_{ij}, i) + \gamma \quad \text{for all true parents } i \in \operatorname{pa}(j),$$

*and*

$$\mathbb{E}[\delta_{i \to j}(q; S_{ij})] \leq \tau_j^{\mathrm{MDL}}(S_{ij}, i) - \gamma \quad \text{for all non-parents } i \notin \operatorname{pa}(j).$$

*If the sub-exponential condition of Thm. 2 holds uniformly with parameters $(\nu, b)$, then the probability of making any decision error at node $j$ satisfies:*

$$\Pr\big(\text{any decision error at node } j\big) \leq 2P_j \exp\left( -cn \min\left\{ \frac{\gamma^2}{\nu^2}, \frac{\gamma}{b} \right\} \right).$$

*This implies that false inclusions and false exclusions vanish exponentially as $n$ increases.*

*Proof.* For any candidate $i$, Thm. 2 implies $\Pr(|\delta_{i \to j} - \mathbb{E}\delta_{i \to j}| \geq \gamma) \leq 2\exp(-cn \min\{\gamma^2/\nu^2, \gamma/b\})$. If $i \in \operatorname{pa}(j)$, a false exclusion occurs only if $\delta_{i \to j} \leq \tau^{\mathrm{MDL}}$, which implies $\delta_{i \to j} - \mathbb{E}\delta_{i \to j} \leq -\gamma$. Similarly, for $i \notin \operatorname{pa}(j)$, a false inclusion occurs only if the deviation is $\geq \gamma$. Applying the union bound over at most $P_j$ candidates yields the claim. $\square$

**Remark (Parametric Add-on).** If $q_{j,S}$ belongs to a parametric family with $d_S$ free parameters trained by Maximum Likelihood Estimation (MLE) on $n$ samples (in contrast to our default prequential usage), one could incorporate a BIC-style penalty term $\frac{1}{2}(d_S - d_{S \setminus \{i\}})\frac{\log n}{n}$ into Eq. (3). Our non-parametric default formulation strictly penalizes the combinatorial search space; the statistical complexity of the predictive model is handled implicitly by the prequential scoring mechanism.

### E.5 BIC CALIBRATION UNDER REGULAR PARAMETRIC CONDITIONS

This subsection provides a classical calibration of PEP's decision rule under regular parametric assumptions. The result is intended for orientation only. It shows that the prequential evidence gain reduces to the usual in-sample likelihood gain up to $o_p((\log n)/n)$ and that, after adding the familiar $\frac{1}{2}\Delta d\frac{\log n}{n}$ term to the gate, the PEP rule recovers a local BIC comparison. The main guarantees of PEP in the paper do not rely on these assumptions and follow instead from the CMI target, regret stability, and prequential concentration.

**Lemma 1** (Reduction to BIC under Regular Parametric Conditions). *Fix a node $j$ and a context $S \subseteq \operatorname{Pred}_\pi(j)$ with $i \in S$, and let $S' = S \setminus \{i\}$. Suppose $q_{j,S}$ and $q_{j,S'}$ are correctly specified, regular parametric conditionals with respective dimensions $d_S$ and $d_{S'}$. Assume i.i.d. data, $K$-fold prequential (sample-splitting) scoring with fixed $K$, and standard regularity conditions (MLE consistency and asymptotic normality, positive-definite Fisher information, and uniform integrability of log-likelihoods). Then,*

$$\delta_{i \to j}(q; S) = \frac{1}{n}\Big( \log L_j(S) - \log L_j(S') \Big) + o_p\left( \frac{\log n}{n} \right), \tag{9}$$

*where $\log L_j(\cdot)$ denotes the in-sample maximized log-likelihood for $X_j$ given the indicated parent set. Define the augmented penalty:*

$$\tau_j^{\mathrm{MDL+BIC}}(S, i) := \tau_j^{\mathrm{MDL}}(S, i) + \frac{1}{2}(d_S - d_{S'})\frac{\log n}{n}, \tag{10}$$

with $\tau_j^{\mathrm{MDL}}(S, i)$ as in Eq. (2). Then the PEP decision rule

$$\delta_{i \to j}(q; S) > \tau_j^{\mathrm{MDL+BIC}}(S, i) \tag{11}$$

is asymptotically equivalent to the local BIC inequality:

$$\underbrace{\left(\log L_j(S) - \log L_j(S')\right) - \frac{1}{2}(d_S - d_{S'}) \log n}_{\Delta \mathrm{BIC}(i \to j; S)} > \log(P_j - k) + \lambda \log(k + 1) + \Omega(n, d) + o_p(1), \tag{12}$$

where $k = |S'|$ and $P_j = |\mathrm{Pred}_\pi(j)|$. In particular, if $P_j - k = 1$ and the combinatorial penalty terms are negligible, the rule reduces asymptotically to $\Delta \mathrm{BIC}(i \to j; S) > 0$.

*Proof.* Let $M_S$ and $M_{S'}$ denote the local parametric families for $S$ and $S'$, with parameters $\theta_S \in \mathbb{R}^{d_S}$ and $\theta_{S'} \in \mathbb{R}^{d_{S'}}$. For a single observation $(x_j^{(s)}, x_S^{(s)})$, let $\ell_S(\theta_S; s) = \log p_{\theta_S}(x_j^{(s)} \mid x_S^{(s)})$ and $\ell_S(\theta_S) = \sum_{s=1}^n \ell_S(\theta_S; s)$. Let $\hat{\theta}_S = \arg\max_{\theta_S} \ell_S(\theta_S)$ be the Maximum Likelihood Estimator (MLE), and similarly for $S'$.

*Step 1 (Prequential–In-sample Alignment).* Let $\{I_k\}_{k=1}^K$ be a fixed $K$-fold partition with $|I_k| = n_k \asymp n/K$. Denote fold-wise MLEs by $\hat{\theta}_S^{(-k)}$ (trained on the complement of $I_k$). Standard M-estimation stability implies $\hat{\theta}_S^{(-k)} - \hat{\theta}_S = O_p(n^{-1})$. A second-order Taylor expansion around $\hat{\theta}_S$, summed over $s \in I_k$ and $k = 1, \ldots, K$, yields:

$$\mathrm{Preq}_S = \sum_{k=1}^K \sum_{s \in I_k} \ell_S(\hat{\theta}_S^{(-k)}; s) = \ell_S(\hat{\theta}_S) + O_p(n^{-1/2}),$$

$$\frac{1}{n}\mathrm{Preq}_S = \frac{1}{n}\ell_S(\hat{\theta}_S) + O_p(n^{-3/2}).$$

An identical relation holds for the subset $S'$.

*Step 2 (Gain Identity).* By the definition of $\delta$ in Eq. (1),

$$\delta_{i \to j}(q; S) = \frac{1}{n}\left(\mathrm{Preq}_S - \mathrm{Preq}_{S'}\right) = \frac{1}{n}\left(\ell_S(\hat{\theta}_S) - \ell_{S'}(\hat{\theta}_{S'})\right) + O_p(n^{-3/2}).$$

This confirms Eq. (9), as $n^{-3/2}$ is negligible compared to $(\log n)/n$.

*Step 3 (Equivalence with Local BIC).* Multiplying Eq. (11) by $n$ and substituting Eq. (9) yields the inequality. Rearranging terms to isolate the BIC components results in Eq. (12), establishing the claim. $\square$

**Scope.** The calibration above relies on fixed-$K$ cross-fitting stability of MLEs and a second-order expansion; it does not invoke Laplace approximations for marginal likelihoods. It demonstrates that prequential (out-of-fold) gains recover the in-sample BIC regime under regular parametric families. However, the default operation of PEP remains model-class agnostic and applies beyond this regime, with guarantees derived from its CMI target, regret stability, and prequential concentration.

# F IMPLEMENTATION DETAILS

All experiments were conducted on a single NVIDIA RTX 6000 GPU. Reported results represent the average over 10 independent runs with distinct random seeds for data generation. In each run, the dataset was partitioned into a training set (context) and a test set (query) to strictly adhere to the prequential principle of out-of-sample evaluation.

Our approach prioritizes a principled design to obviate per-dataset tuning. For the PEP framework, we fixed the structural scaling factor at $\eta = 1$, consistent with the theoretical derivation in § 3.3. The predictive component was instantiated using the pre-trained `TabPFNv2` (Hollmann et al., 2025a) model without fine-tuning.

For the comparative experiments involving alternative predictors (Random Forest, XGBoost, Cat-Boost, LightGBM), we employed the `AutoGluon` framework[3] (Erickson et al., 2020) to ensure a standardized implementation. We utilized the default hyperparameter settings provided by Auto-Gluon to avoid manual tuning bias and applied Platt scaling to the outputs of these models to ensure probability calibration.

## F.1 BENCHMARK DATASETS

To ensure a rigorous evaluation, we designed two distinct experimental settings tailored to the specific goals of each analysis:

- **Main Performance Benchmarks (SynER and SynSF):** We configured the functional relationships to be fully non-linear ($\rho_{\text{lin}} = 0.0$). This setting ensures a fair comparison with score-based ordering methods (e.g., SCORE, DAS, NoGAM), which typically rely on non-linear identifiability assumptions.

- **Misspecification Stress Tests:** Since this suite includes a scenario specifically designed for purely linear relationships (LiNGAM), we established the baseline (vanilla) environment as a mixed setting with a linearity probability of $\rho_{\text{lin}} = 0.5$. This dual setup allows us to validate the robustness of our method under both idealized non-linear conditions and more general, heterogeneous environments.

**Synthetic Dataset Generation Details.** All synthetic datasets were generated via a two-step process: (1) sampling a ground truth Directed Acyclic Graph (DAG) from a random graph model, and (2) sampling data from a Structural Equation Model (SEM) defined by that DAG. Unless stated otherwise (e.g., in scalability experiments), the primary comparative benchmarks employ a default configuration with $d = 10$ nodes, $n = 2000$ samples, and dense graphs having an expected number of edges equal to $4d$.

- **Erdös-Rényi (ER) Graphs:** The ER model (Erdős & Rényi, 1960) generates homogeneous graph structures. For a given number of nodes $d$, each possible undirected edge is included with a fixed, uniform probability $p$. To enforce acyclicity, we first establish a random permutation of the nodes to define a topological order and then orient the selected edges to be consistent with this order. The resulting graphs are characterized by a degree distribution that approximates a Poisson distribution.

- **Scale-Free (SF) Graphs:** The SF model (Bollobás et al., 2003) generates heterogeneous structures that mimic real-world networks. We utilize the Barabási-Albert model, which employs a preferential attachment mechanism. The graph grows iteratively: at each step, a new node is added and connected to existing nodes with a probability proportional to their current degree. This "rich-get-richer" dynamic results in a power-law degree distribution, characterized by a few highly connected hubs and many sparsely connected nodes. Similar to the ER model, edge orientations are determined by a random topological order.

**Real-World Benchmark Details.** To assess performance in practical scenarios, we utilized two established real-world benchmark datasets:

- **Sachs:** The Sachs dataset (Sachs et al., 2005) is a standard benchmark derived from a protein-signaling network in human primary T cells ($n = 853$, $d = 11$). The ground truth causal graph, established through expert knowledge and interventional experiments, contains 20 edges. This dataset evaluates the ability to recover known biological pathways from observational flow cytometry data.

- **SynTReN:** The SynTReN (Synthetic Transcriptional Regulatory Network) dataset (Van den Bulcke et al., 2006) is a pseudo-real-world benchmark that simulates gene expression data. The underlying network structure is extracted from the *E. coli* transcriptional regulatory network (not random), while the observational data is generated using a kinetic model that simulates transcription and translation dynamics. For our experiments, we use a version with $d = 20$ nodes (genes) and $n = 500$ samples. This dataset challenges algorithms with realistic, non-random graph structures and complex noise profiles.

---

[3] https://github.com/autogluon/autogluon

**Misspecified Scenario Details.** To rigorously evaluate robustness, we generated synthetic datasets under six scenarios designed to systematically violate core causal discovery assumptions, following the methodology of Montagna et al. (2023a). The parameters were set as follows: confounder probability $\rho = 0.2$, signal-to-noise ratio $\gamma = 0.8$ for measurement error, unfaithfulness probability $p_{\text{unfaithful}} = 0.3$, and an exponent of 3.0 for post-nonlinear transformations.

- **Latent Confounders:** Violates causal sufficiency. For randomly selected pairs $(X_i, X_j)$ without a direct edge, we introduce a latent confounder $C$. The generation process becomes $X_i = f_i(\text{pa}(i) \cup \{C\}) + \epsilon_i$ and $X_j = f_j(\text{pa}(j) \cup \{C\}) + \epsilon_j$, inducing spurious correlations that test the algorithm's ability to avoid false positives.

- **Measurement Error:** Violates the assumption of error-free measurement. Observed data $\tilde{X}$ is generated by adding independent Gaussian noise to the true values $X$: $\tilde{X}_i := X_i + \eta_i$, where $\eta_i \sim \mathcal{N}(0, \sigma_\eta^2)$. This tests resilience to data corruption.

- **Unfaithful Distributions:** Violates the faithfulness assumption. We create cancelling paths by adding a direct edge $X_i \rightarrow X_k$ to a path $X_i \rightarrow X_j \rightarrow X_k$. The parameters are tuned such that the causal effects cancel out, rendering $X_i$ and $X_k$ marginally independent ($X_i \perp\!\!\!\perp X_k$). This tests the ability to recover true edges despite masked statistical signals.

- **Autoregressive Model (Non-i.i.d.):** Violates the i.i.d. assumption. We introduce temporal dependency via an AR(1) model: $x^{(s)} = Ax^{(s-1)} + \epsilon^{(s)}$, where $A$ is the adjacency matrix. This tests robustness to temporal correlations.

- **Post-Nonlinear (PNL) Models:** Violates the additivity assumption. A non-linear distortion $g_j$ is applied to the entire mechanism: $X_j = g_j(\sum_{k \in \text{pa}(j)} f_{j,k}(X_k) + \epsilon_j)$. This creates complex non-additive interactions, testing model flexibility.

- **Linear Non-Gaussian Acyclic Model (LiNGAM):** Violates the Gaussian noise assumption required by some score-based methods. Data is generated from a linear SEM with non-Gaussian (uniform) noise $\epsilon_j$. This tests the algorithm's reliance on Gaussianity for identifiability.

## F.2 Baseline Selection

We benchmark PEP against a comprehensive suite of state-of-the-art ordering-based causal discovery algorithms. While **DAS** was evaluated alongside other methods, we report its results primarily in this appendix. Since DAS shares the exact same ordering mechanism as **SCORE**, applying a deterministic pruning module like PEP yields identical structural results for both backbones. Therefore, to avoid redundancy, we utilize SCORE as the representative baseline for this family of variance-based algorithms in the main text.

We utilized the implementations for CAM, SCORE, DAS, and NoGAM from the `dodiscover` package[4]. For DiffAN and CaPS, we used the authors' original implementations[5]. The specific characteristics of each baseline are as follows:

- **CAM**: The Causal Additive Models algorithm (Bühlmann et al., 2014) decouples discovery into two stages: likelihood-based ordering and GAM-based pruning. It estimates the topological order by maximizing the restricted likelihood of the additive SEM via greedy search. For pruning, it fits a Generalized Additive Model (GAM) for each node $X_j$ against its predecessors and tests the null hypothesis $H_0 : f_{j,k}(\cdot) \equiv 0$ for each parent candidate $X_k$. Edges are retained based on the statistical significance (p-value) of the contribution.

- **SCORE**: This algorithm (Rolland et al., 2022) identifies the topological order by recursively finding leaf nodes. Under non-linear assumptions, a node $X_j$ is a leaf if and only if the variance of the diagonal of the score Jacobian is zero. The score function $s(x) = \nabla_x \log p(x)$ is estimated using a Stein gradient estimator. The leaf identification criterion is:

$$\hat{j} = \arg\min_j \text{Var}\left[\frac{\partial s_j(x)}{\partial x_j}\right].$$

---

[4] https://github.com/py-why/dodiscover
[5] https://github.com/vios-s/DiffAN, https://github.com/E2real/CaPS

By default, SCORE employs CAM pruning on the fully connected DAG derived from the estimated order.

- **DAS**: The Discovery At Scale algorithm (Montagna et al., 2023b) utilizes the same variance-based ordering criterion as SCORE. Its primary innovation lies in an intermediate pruning stage that uses off-diagonal elements of the score Jacobian. It performs an initial, computationally efficient edge selection based on $\mathbb{E}[|\partial_{X_k} s_j(x)|] \neq 0 \iff X_k \in \mathrm{pa}(j)$. This step reduces the candidate set for the final pruning stage, which typically defaults to CAM pruning to refine the graph.

- **NoGAM**: The NoGAM algorithm (Montagna et al., 2023c) generalizes score-based ordering to arbitrary additive noise models. It identifies leaf nodes by minimizing the mean squared error of a score prediction derived from estimated noise residuals $R_j$. The criterion is formulated as:

$$\hat{j} = \arg\min_j \mathbb{E}\left[(\mathbb{E}[s_j(X) \mid R_j] - s_j(X))^2\right].$$

The score function is approximated via score matching based on Stein's identity. Like other score-based methods, it relies on post-processing (e.g., CAM pruning) to obtain the final DAG.

- **DiffAN**: This algorithm (Sanchez et al., 2023) adopts the variance-based leaf identification criterion of SCORE but introduces a scalable score estimation method. Instead of kernel-based estimation, DiffAN trains a probabilistic diffusion model to approximate the score and its Jacobian via backpropagation. It employs the *deciduous score* update to efficiently handle iterative leaf removal without retraining. The final graph is obtained via standard post-processing pruning.

- **CaPS**: The Causal Discovery with Parent Score algorithm (Xu et al., 2024) proposes an ordering criterion robust to mixed linear and non-linear settings. It identifies leaf nodes by maximizing the expectation, rather than the variance, of the score Jacobian diagonal:

$$\hat{j} = \arg\max_j \left(\mathrm{diag}\left(\mathbb{E}\left[\frac{\partial s(x)}{\partial x}\right]\right)\right).$$

CaPS utilizes a "parent score" for efficient pre-pruning of weak edges and supplementation of strong edges, reducing the computational burden on the final CAM pruning step.

### F.3 EVALUATION METRICS

We evaluate the accuracy of the recovered graph structures using a suite of standard metrics. Let TP (True Positives) denote the number of correctly identified edges, FP (False Positives) the number of incorrectly identified edges, FN (False Negatives) the number of missed true edges, and R the number of edges with a reversed direction.

- **Structural Hamming Distance (SHD):** The SHD measures the overall structural dissimilarity between the estimated graph and the ground truth graph. It is defined as the total number of edge operations (additions, deletions, or reversals) required to make the two graphs identical:

$$\mathrm{SHD} = \mathrm{FP} + \mathrm{FN} + \mathrm{R}.$$

A lower SHD indicates a more accurate structural recovery.

**Normalized and Inverted SHD (SHD$^\dagger$):** For visualization purposes (e.g., in radar charts where larger areas imply better performance), we report a normalized and inverted version of SHD. Since the maximum possible SHD for a graph with $d$ nodes is bounded by the total number of possible edges $d(d-1)$, we define:

$$\mathrm{SHD}^\dagger = 1 - \frac{\mathrm{SHD}}{d(d-1)}.$$

Here, $\mathrm{SHD}^\dagger \in [0, 1]$, where 1 indicates a perfect match.

- **Structural Intervention Distance (SID):** The SID is a causally-informed metric that quantifies the number of downstream errors in interventional reasoning resulting from the

estimated graph. It counts the pairs of variables $(i, j)$ for which the set of causal paths from $i$ to $j$ is incorrectly estimated. A lower SID indicates that the graph is more faithful for predicting intervention effects.

**Normalized and Inverted SID (SID$^\dagger$):** Similar to SHD, we normalize SID by its maximum possible value $d(d-1)$ and invert it to align with accuracy metrics:

$$\text{SID}^\dagger = 1 - \frac{\text{SID}}{d(d-1)}.$$

A value of SID$^\dagger$ closer to 1 signifies better causal reasoning capability.

- **Precision, Recall, and F1 Score:** These metrics assess edge discovery accuracy by treating the problem as a binary classification task for each potential edge.

    – **Precision** measures the fraction of predicted edges that are correct:

    $$\text{Precision} = \frac{\text{TP}}{\text{TP} + \text{FP}}.$$

    – **Recall** (True Positive Rate) measures the fraction of true edges correctly identified:

    $$\text{Recall} = \frac{\text{TP}}{\text{TP} + \text{FN}}.$$

    – The **F1 Score** is the harmonic mean of Precision and Recall:

    $$\text{F1 Score} = \frac{2 \cdot \text{Precision} \cdot \text{Recall}}{\text{Precision} + \text{Recall}}.$$

    – **Note on Reversed Edges:** We treat reversed edges (R) as a distinct error type. For metrics like the False Discovery Rate (FDR) or False Positive Rate (FPR), reversed edges are included in the numerator alongside false positives (e.g., FPR = (R + FP)/(TN + FP)). We adopt this strict convention because a reversed edge, while identifying an adjacency, represents a fundamentally incorrect causal claim and should be penalized as a false discovery.

## G ADDITIONAL EXPERIMENTAL RESULTS

**Detailed Numerical Results.** This section provides the precise quantitative data corresponding to the visualizations presented in the main text. We report the mean and standard deviation for all experiments in tabular form to ensure transparency and reproducibility. Specifically, the numerical results for the structural penalty ablation study (Fig. 5) and the misspecification stress tests (Fig. 3) are detailed in Table G.1, Table G.2, and Table G.3, respectively.

Table G.1: **Detailed numerical results for the ablation study on the structural penalty scaling factor $\eta$.** All experiments utilize the **SCORE** ordering backbone. The table is split into two panels for readability: low-dimensional graphs ($d = 10, 30$) on top and high-dimensional graphs ($d = 50, 100$) below. While weaker regularization ($\eta < 1.0$) suffices for small $d$, the theoretical baseline ($\eta = 1.0$) is essential for performance in high-dimensional regimes.

| Factor $\eta$ | $d = 10$ | | | $d = 30$ | | |
|---|---|---|---|---|---|---|
| | SHD $\downarrow$ | SID $\downarrow$ | F1 $\uparrow$ | SHD $\downarrow$ | SID $\downarrow$ | F1 $\uparrow$ |
| 0.0 (No penalty) | $7.1_{(3.0)}$ | $\mathbf{3.7}_{(4.8)}$ | $0.90_{(0.05)}$ | $143.4_{(10.2)}$ | $224.6_{(51.9)}$ | $0.52_{(0.02)}$ |
| 0.25 | $\mathbf{5.1}_{(3.6)}$ | $5.2_{(4.8)}$ | $\mathbf{0.92}_{(0.06)}$ | $77.0_{(13.7)}$ | $261.4_{(44.2)}$ | $0.65_{(0.04)}$ |
| 0.50 | $5.2_{(3.4)}$ | $6.7_{(4.7)}$ | $0.91_{(0.06)}$ | $61.8_{(14.4)}$ | $\mathbf{259.7}_{(50.1)}$ | $0.69_{(0.05)}$ |
| 0.75 | $5.4_{(3.1)}$ | $7.7_{(5.3)}$ | $0.91_{(0.05)}$ | $55.7_{(13.2)}$ | $269.8_{(47.4)}$ | $\mathbf{0.70}_{(0.06)}$ |
| **1.0 (Theoretical)** | $5.4_{(3.4)}$ | $8.5_{(5.9)}$ | $0.91_{(0.06)}$ | $\mathbf{55.3}_{(12.9)}$ | $269.3_{(48.6)}$ | $\mathbf{0.70}_{(0.07)}$ |

| Factor $\eta$ | $d = 50$ | | | $d = 100$ | | |
|---|---|---|---|---|---|---|
| | SHD $\downarrow$ | SID $\downarrow$ | F1 $\uparrow$ | SHD $\downarrow$ | SID $\downarrow$ | F1 $\uparrow$ |
| 0.0 (No penalty) | $364.1_{(37.3)}$ | $900.9_{(149.2)}$ | $0.38_{(0.02)}$ | $1264.6_{(153.8)}$ | $4290.1_{(361.8)}$ | $0.25_{(0.02)}$ |
| 0.25 | $166.9_{(19.6)}$ | $971.7_{(170.2)}$ | $0.54_{(0.05)}$ | $469.7_{(76.3)}$ | $4554.0_{(403.1)}$ | $0.45_{(0.04)}$ |
| 0.50 | $127.6_{(19.1)}$ | $\mathbf{969.2}_{(158.6)}$ | $0.60_{(0.06)}$ | $300.9_{(38.6)}$ | $\mathbf{4474.6}_{(433.5)}$ | $0.54_{(0.03)}$ |
| 0.75 | $115.1_{(20.3)}$ | $973.3_{(148.1)}$ | $\mathbf{0.62}_{(0.06)}$ | $262.0_{(30.4)}$ | $4514.3_{(384.7)}$ | $0.56_{(0.04)}$ |
| **1.0 (Theoretical)** | $\mathbf{114.9}_{(21.2)}$ | $974.3_{(148.5)}$ | $\mathbf{0.62}_{(0.07)}$ | $\mathbf{246.5}_{(23.3)}$ | $4478.4_{(393.4)}$ | $\mathbf{0.57}_{(0.03)}$ |

Table G.2: **Detailed scenario comparison on SynER** ($d = 10$). Rows are grouped by Scenario and Ordering Backbone, while columns represent the evaluation metrics. Standard deviations are reported in subscripts. **Bold** indicates the better performance between CAM-pruning (Base) and PEP.

| Scenario | Ordering | Pruning | SHD ↓ | SID ↓ | F1 ↑ | Precision ↑ | Recall ↑ |
|---|---|---|---|---|---|---|---|
| **PNL** | CAM | Base | $21.40_{(4.40)}$ | $40.10_{(10.28)}$ | $0.61_{(0.09)}$ | $\mathbf{0.94}_{(0.04)}$ | $0.43_{(0.08)}$ |
| | | PEP | $\mathbf{18.00}_{(3.77)}$ | $\mathbf{39.40}_{(11.57)}$ | $\mathbf{0.68}_{(0.08)}$ | $0.76_{(0.06)}$ | $\mathbf{0.66}_{(0.08)}$ |
| | SCORE | Base | $21.20_{(4.18)}$ | $38.20_{(8.68)}$ | $0.62_{(0.09)}$ | $\mathbf{0.99}_{(0.02)}$ | $0.45_{(0.08)}$ |
| | | PEP | $\mathbf{8.60}_{(2.54)}$ | $\mathbf{18.50}_{(6.66)}$ | $\mathbf{0.86}_{(0.06)}$ | $0.94_{(0.03)}$ | $\mathbf{0.84}_{(0.06)}$ |
| | NoGAM | Base | $20.90_{(4.10)}$ | $38.60_{(9.55)}$ | $0.62_{(0.09)}$ | $\mathbf{0.99}_{(0.02)}$ | $0.45_{(0.08)}$ |
| | | PEP | $\mathbf{9.70}_{(3.46)}$ | $\mathbf{20.50}_{(7.40)}$ | $\mathbf{0.84}_{(0.07)}$ | $0.94_{(0.03)}$ | $\mathbf{0.81}_{(0.07)}$ |
| | DiffAN | Base | $22.40_{(5.28)}$ | $56.00_{(12.85)}$ | $0.57_{(0.12)}$ | $\mathbf{0.84}_{(0.08)}$ | $0.43_{(0.11)}$ |
| | | PEP | $\mathbf{18.10}_{(4.87)}$ | $\mathbf{49.50}_{(8.90)}$ | $\mathbf{0.67}_{(0.08)}$ | $0.81_{(0.06)}$ | $\mathbf{0.64}_{(0.07)}$ |
| | CaPS | Base | $19.50_{(4.37)}$ | $35.40_{(8.50)}$ | $0.64_{(0.09)}$ | $0.96_{(0.04)}$ | $0.51_{(0.09)}$ |
| | | PEP | $\mathbf{7.00}_{(2.22)}$ | $\mathbf{15.20}_{(6.32)}$ | $\mathbf{0.89}_{(0.05)}$ | $\mathbf{0.97}_{(0.02)}$ | $\mathbf{0.87}_{(0.05)}$ |
| **LiNGAM** | CAM | Base | $10.20_{(2.56)}$ | $24.20_{(7.19)}$ | $0.78_{(0.08)}$ | $\mathbf{0.93}_{(0.05)}$ | $0.66_{(0.10)}$ |
| | | PEP | $\mathbf{8.60}_{(2.32)}$ | $\mathbf{22.90}_{(7.69)}$ | $\mathbf{0.86}_{(0.06)}$ | $0.84_{(0.06)}$ | $\mathbf{0.88}_{(0.06)}$ |
| | SCORE | Base | $9.90_{(2.58)}$ | $21.60_{(6.52)}$ | $0.80_{(0.08)}$ | $\mathbf{0.97}_{(0.03)}$ | $0.69_{(0.11)}$ |
| | | PEP | $\mathbf{3.80}_{(2.18)}$ | $\mathbf{9.10}_{(4.21)}$ | $\mathbf{0.93}_{(0.04)}$ | $0.95_{(0.04)}$ | $\mathbf{0.96}_{(0.04)}$ |
| | NoGAM | Base | $9.90_{(2.37)}$ | $21.40_{(6.33)}$ | $0.81_{(0.07)}$ | $\mathbf{0.98}_{(0.03)}$ | $0.69_{(0.11)}$ |
| | | PEP | $\mathbf{3.90}_{(2.14)}$ | $\mathbf{9.60}_{(4.42)}$ | $\mathbf{0.91}_{(0.05)}$ | $0.94_{(0.04)}$ | $\mathbf{0.94}_{(0.04)}$ |
| | DiffAN | Base | $12.00_{(3.25)}$ | $33.10_{(10.41)}$ | $0.72_{(0.09)}$ | $\mathbf{0.79}_{(0.08)}$ | $0.64_{(0.12)}$ |
| | | PEP | $\mathbf{8.70}_{(2.68)}$ | $\mathbf{30.30}_{(10.57)}$ | $\mathbf{0.80}_{(0.06)}$ | $\mathbf{0.79}_{(0.06)}$ | $\mathbf{0.83}_{(0.06)}$ |
| | CaPS | Base | $7.90_{(3.26)}$ | $18.40_{(6.35)}$ | $0.83_{(0.06)}$ | $0.95_{(0.04)}$ | $0.74_{(0.10)}$ |
| | | PEP | $\mathbf{2.80}_{(1.54)}$ | $\mathbf{7.80}_{(4.50)}$ | $\mathbf{0.94}_{(0.03)}$ | $\mathbf{0.96}_{(0.03)}$ | $\mathbf{0.96}_{(0.04)}$ |
| **Confounded** | CAM | Base | $25.20_{(4.60)}$ | $49.50_{(12.41)}$ | $0.56_{(0.09)}$ | $\mathbf{0.90}_{(0.05)}$ | $0.44_{(0.09)}$ |
| | | PEP | $\mathbf{22.40}_{(4.10)}$ | $\mathbf{49.30}_{(14.97)}$ | $\mathbf{0.62}_{(0.08)}$ | $0.72_{(0.06)}$ | $\mathbf{0.74}_{(0.07)}$ |
| | SCORE | Base | $24.60_{(4.59)}$ | $45.20_{(11.03)}$ | $0.57_{(0.10)}$ | $\mathbf{0.96}_{(0.03)}$ | $0.45_{(0.10)}$ |
| | | PEP | $\mathbf{10.00}_{(2.63)}$ | $\mathbf{23.80}_{(8.61)}$ | $\mathbf{0.83}_{(0.06)}$ | $0.93_{(0.04)}$ | $\mathbf{0.89}_{(0.06)}$ |
| | NoGAM | Base | $24.50_{(4.31)}$ | $45.30_{(10.31)}$ | $0.58_{(0.10)}$ | $\mathbf{0.97}_{(0.03)}$ | $0.47_{(0.10)}$ |
| | | PEP | $\mathbf{11.70}_{(3.01)}$ | $\mathbf{27.30}_{(9.11)}$ | $\mathbf{0.81}_{(0.07)}$ | $0.93_{(0.04)}$ | $\mathbf{0.86}_{(0.07)}$ |
| | DiffAN | Base | $25.80_{(5.64)}$ | $65.10_{(14.66)}$ | $0.52_{(0.12)}$ | $\mathbf{0.82}_{(0.08)}$ | $0.46_{(0.12)}$ |
| | | PEP | $\mathbf{22.10}_{(5.13)}$ | $\mathbf{58.40}_{(11.49)}$ | $\mathbf{0.63}_{(0.08)}$ | $0.79_{(0.06)}$ | $\mathbf{0.68}_{(0.07)}$ |
| | CaPS | Base | $22.80_{(4.82)}$ | $41.20_{(10.08)}$ | $0.62_{(0.10)}$ | $0.94_{(0.04)}$ | $0.53_{(0.11)}$ |
| | | PEP | $\mathbf{9.10}_{(2.35)}$ | $\mathbf{22.50}_{(8.53)}$ | $\mathbf{0.86}_{(0.05)}$ | $\mathbf{0.96}_{(0.03)}$ | $\mathbf{0.88}_{(0.05)}$ |
| **Measure-Err** | CAM | Base | $20.00_{(4.43)}$ | $52.50_{(12.19)}$ | $0.51_{(0.11)}$ | $\mathbf{0.88}_{(0.05)}$ | $0.39_{(0.10)}$ |
| | | PEP | $\mathbf{18.30}_{(4.10)}$ | $\mathbf{49.70}_{(12.33)}$ | $\mathbf{0.57}_{(0.10)}$ | $0.66_{(0.08)}$ | $\mathbf{0.51}_{(0.13)}$ |
| | SCORE | Base | $19.80_{(4.24)}$ | $49.50_{(10.66)}$ | $0.52_{(0.10)}$ | $\mathbf{0.94}_{(0.03)}$ | $0.40_{(0.10)}$ |
| | | PEP | $\mathbf{8.40}_{(2.31)}$ | $\mathbf{24.80}_{(8.70)}$ | $\mathbf{0.79}_{(0.07)}$ | $0.91_{(0.05)}$ | $\mathbf{0.75}_{(0.08)}$ |
| | NoGAM | Base | $19.50_{(4.30)}$ | $49.90_{(12.28)}$ | $0.52_{(0.10)}$ | $\mathbf{0.95}_{(0.03)}$ | $0.41_{(0.10)}$ |
| | | PEP | $\mathbf{9.40}_{(3.40)}$ | $\mathbf{27.60}_{(8.65)}$ | $\mathbf{0.76}_{(0.07)}$ | $0.90_{(0.05)}$ | $\mathbf{0.73}_{(0.08)}$ |
| | DiffAN | Base | $21.70_{(4.83)}$ | $63.89_{(8.07)}$ | $0.47_{(0.12)}$ | $\mathbf{0.78}_{(0.07)}$ | $0.41_{(0.11)}$ |
| | | PEP | $\mathbf{18.70}_{(4.81)}$ | $\mathbf{60.00}_{(19.43)}$ | $\mathbf{0.59}_{(0.12)}$ | $0.67_{(0.12)}$ | $\mathbf{0.55}_{(0.20)}$ |
| | CaPS | Base | $18.50_{(4.29)}$ | $44.80_{(10.08)}$ | $0.60_{(0.12)}$ | $0.92_{(0.04)}$ | $0.48_{(0.12)}$ |
| | | PEP | $\mathbf{7.40}_{(2.06)}$ | $\mathbf{48.30}_{(9.78)}$ | $\mathbf{0.81}_{(0.08)}$ | $0.89_{(0.08)}$ | $\mathbf{0.73}_{(0.14)}$ |
| **Non-i.i.d** | CAM | Base | $9.70_{(2.33)}$ | $24.70_{(7.56)}$ | $0.78_{(0.08)}$ | $\mathbf{0.93}_{(0.05)}$ | $0.66_{(0.10)}$ |
| | | PEP | $\mathbf{8.40}_{(2.50)}$ | $\mathbf{23.60}_{(7.71)}$ | $\mathbf{0.86}_{(0.07)}$ | $0.84_{(0.06)}$ | $\mathbf{0.88}_{(0.06)}$ |
| | SCORE | Base | $9.40_{(2.39)}$ | $22.00_{(6.80)}$ | $0.80_{(0.08)}$ | $\mathbf{0.97}_{(0.03)}$ | $0.68_{(0.10)}$ |
| | | PEP | $\mathbf{3.60}_{(2.13)}$ | $\mathbf{9.20}_{(4.18)}$ | $\mathbf{0.93}_{(0.04)}$ | $0.95_{(0.04)}$ | $\mathbf{0.96}_{(0.04)}$ |
| | NoGAM | Base | $9.30_{(2.33)}$ | $21.80_{(6.42)}$ | $0.81_{(0.07)}$ | $\mathbf{0.98}_{(0.03)}$ | $0.69_{(0.10)}$ |
| | | PEP | $\mathbf{3.80}_{(2.14)}$ | $\mathbf{9.60}_{(4.53)}$ | $\mathbf{0.91}_{(0.05)}$ | $0.94_{(0.04)}$ | $\mathbf{0.94}_{(0.04)}$ |
| | DiffAN | Base | $11.20_{(3.02)}$ | $32.40_{(10.56)}$ | $0.72_{(0.08)}$ | $\mathbf{0.79}_{(0.07)}$ | $0.64_{(0.11)}$ |
| | | PEP | $\mathbf{8.20}_{(2.65)}$ | $\mathbf{29.80}_{(10.84)}$ | $\mathbf{0.80}_{(0.06)}$ | $\mathbf{0.79}_{(0.06)}$ | $\mathbf{0.83}_{(0.06)}$ |
| | CaPS | Base | $7.40_{(2.67)}$ | $18.10_{(6.17)}$ | $0.83_{(0.06)}$ | $0.95_{(0.04)}$ | $0.73_{(0.10)}$ |
| | | PEP | $\mathbf{2.90}_{(1.46)}$ | $\mathbf{8.10}_{(4.64)}$ | $\mathbf{0.94}_{(0.03)}$ | $\mathbf{0.96}_{(0.03)}$ | $\mathbf{0.96}_{(0.04)}$ |
| **Unfaithful** | CAM | Base | $22.50_{(3.57)}$ | $66.00_{(11.04)}$ | $0.48_{(0.07)}$ | $\mathbf{0.86}_{(0.07)}$ | $0.35_{(0.08)}$ |
| | | PEP | $\mathbf{19.70}_{(3.08)}$ | $\mathbf{60.80}_{(12.44)}$ | $\mathbf{0.55}_{(0.07)}$ | $0.68_{(0.07)}$ | $\mathbf{0.64}_{(0.07)}$ |
| | SCORE | Base | $22.10_{(3.44)}$ | $60.90_{(10.38)}$ | $0.50_{(0.07)}$ | $\mathbf{0.93}_{(0.04)}$ | $0.36_{(0.08)}$ |
| | | PEP | $\mathbf{9.70}_{(2.45)}$ | $\mathbf{29.40}_{(9.20)}$ | $\mathbf{0.79}_{(0.06)}$ | $0.92_{(0.05)}$ | $\mathbf{0.83}_{(0.06)}$ |
| | NoGAM | Base | $22.20_{(3.64)}$ | $59.80_{(11.29)}$ | $0.50_{(0.07)}$ | $\mathbf{0.94}_{(0.04)}$ | $0.37_{(0.08)}$ |
| | | PEP | $\mathbf{11.10}_{(3.12)}$ | $\mathbf{33.60}_{(9.69)}$ | $\mathbf{0.77}_{(0.07)}$ | $0.91_{(0.05)}$ | $\mathbf{0.82}_{(0.07)}$ |
| | DiffAN | Base | $24.10_{(4.37)}$ | $83.10_{(13.72)}$ | $0.46_{(0.10)}$ | $\mathbf{0.77}_{(0.07)}$ | $0.36_{(0.11)}$ |
| | | PEP | $\mathbf{19.30}_{(3.90)}$ | $\mathbf{72.20}_{(12.47)}$ | $\mathbf{0.57}_{(0.08)}$ | $0.76_{(0.06)}$ | $\mathbf{0.62}_{(0.07)}$ |
| | CaPS | Base | $20.60_{(3.61)}$ | $55.50_{(9.33)}$ | $0.57_{(0.09)}$ | $0.92_{(0.05)}$ | $0.45_{(0.10)}$ |
| | | PEP | $\mathbf{8.50}_{(2.14)}$ | $\mathbf{27.60}_{(8.65)}$ | $\mathbf{0.82}_{(0.05)}$ | $\mathbf{0.94}_{(0.04)}$ | $\mathbf{0.85}_{(0.06)}$ |

Table G.3: **Detailed scenario comparison on SynSF** ($d = 10$). Rows are grouped by Scenario and Ordering Backbone, while columns represent the evaluation metrics. Standard deviations are reported in subscripts. **Bold** indicates the better performance between CAM-pruning (Base) and PEP.

| Scenario | Ordering | Pruning | SHD ↓ | SID ↓ | F1 ↑ | Precision ↑ | Recall ↑ |
|---|---|---|---|---|---|---|---|
| PNL | CAM | Base | $18.00_{(5.40)}$ | $58.40_{(18.40)}$ | $0.44_{(0.18)}$ | $\mathbf{0.56}_{(0.23)}$ | $0.37_{(0.15)}$ |
| | | PEP | $\mathbf{13.00}_{(8.04)}$ | $\mathbf{36.75}_{(20.25)}$ | $\mathbf{0.67}_{(0.21)}$ | $0.62_{(0.25)}$ | $\mathbf{0.75}_{(0.16)}$ |
| | SCORE | Base | $14.70_{(3.16)}$ | $41.00_{(8.26)}$ | $0.59_{(0.09)}$ | $\mathbf{0.77}_{(0.12)}$ | $0.48_{(0.11)}$ |
| | | PEP | $\mathbf{12.00}_{(1.58)}$ | $\mathbf{22.40}_{(7.77)}$ | $\mathbf{0.73}_{(0.05)}$ | $0.64_{(0.04)}$ | $\mathbf{0.86}_{(0.06)}$ |
| | NoGAM | Base | $13.90_{(3.51)}$ | $36.80_{(11.17)}$ | $0.61_{(0.11)}$ | $\mathbf{0.80}_{(0.14)}$ | $0.50_{(0.12)}$ |
| | | PEP | $\mathbf{10.60}_{(2.30)}$ | $\mathbf{19.00}_{(12.41)}$ | $\mathbf{0.76}_{(0.06)}$ | $0.67_{(0.06)}$ | $\mathbf{0.88}_{(0.07)}$ |
| | DiffAN | Base | $14.33_{(4.27)}$ | $50.67_{(16.03)}$ | $0.58_{(0.12)}$ | $\mathbf{0.69}_{(0.15)}$ | $0.51_{(0.11)}$ |
| | | PEP | $\mathbf{11.00}_{(6.08)}$ | $\mathbf{26.00}_{(7.00)}$ | $\mathbf{0.73}_{(0.12)}$ | $0.66_{(0.14)}$ | $\mathbf{0.83}_{(0.07)}$ |
| | CaPS | Base | $13.20_{(3.26)}$ | $38.70_{(7.82)}$ | $0.64_{(0.08)}$ | $0.75_{(0.11)}$ | $0.56_{(0.09)}$ |
| | | PEP | $\mathbf{7.50}_{(2.07)}$ | $\mathbf{27.75}_{(8.01)}$ | $\mathbf{0.80}_{(0.05)}$ | $\mathbf{0.79}_{(0.09)}$ | $\mathbf{0.80}_{(0.05)}$ |
| LiNGAM | CAM | Base | $28.83_{(1.83)}$ | $73.50_{(8.69)}$ | $0.19_{(0.04)}$ | $0.17_{(0.03)}$ | $0.22_{(0.06)}$ |
| | | PEP | $\mathbf{28.33}_{(1.15)}$ | $\mathbf{65.00}_{(5.29)}$ | $\mathbf{0.26}_{(0.05)}$ | $\mathbf{0.21}_{(0.04)}$ | $\mathbf{0.32}_{(0.06)}$ |
| | SCORE | Base | $\mathbf{4.00}_{(3.23)}$ | $15.20_{(14.05)}$ | $\mathbf{0.89}_{(0.09)}$ | $\mathbf{0.89}_{(0.09)}$ | $0.90_{(0.09)}$ |
| | | PEP | $6.40_{(4.56)}$ | $\mathbf{6.80}_{(10.43)}$ | $0.86_{(0.10)}$ | $0.79_{(0.13)}$ | $\mathbf{0.95}_{(0.05)}$ |
| | NoGAM | Base | $\mathbf{4.10}_{(2.51)}$ | $15.90_{(12.25)}$ | $0.89_{(0.07)}$ | $\mathbf{0.90}_{(0.08)}$ | $0.88_{(0.07)}$ |
| | | PEP | $4.60_{(2.30)}$ | $\mathbf{5.20}_{(5.40)}$ | $0.89_{(0.06)}$ | $0.84_{(0.08)}$ | $\mathbf{0.96}_{(0.03)}$ |
| | DiffAN | Base | $19.00_{(4.57)}$ | $57.50_{(6.41)}$ | $0.51_{(0.09)}$ | $0.46_{(0.10)}$ | $0.59_{(0.08)}$ |
| | | PEP | $\mathbf{18.60}_{(5.13)}$ | $\mathbf{40.80}_{(11.63)}$ | $\mathbf{0.58}_{(0.10)}$ | $\mathbf{0.49}_{(0.11)}$ | $\mathbf{0.72}_{(0.08)}$ |
| | CaPS | Base | $4.20_{(2.94)}$ | $14.70_{(12.68)}$ | $0.89_{(0.08)}$ | $\mathbf{0.90}_{(0.10)}$ | $0.89_{(0.07)}$ |
| | | PEP | $\mathbf{3.44}_{(3.64)}$ | $\mathbf{8.00}_{(9.11)}$ | $\mathbf{0.92}_{(0.08)}$ | $0.89_{(0.11)}$ | $\mathbf{0.94}_{(0.06)}$ |
| Confounded | CAM | Base | $17.20_{(5.05)}$ | $52.00_{(11.55)}$ | $0.53_{(0.15)}$ | $\mathbf{0.57}_{(0.18)}$ | $0.50_{(0.14)}$ |
| | | PEP | $\mathbf{16.00}_{(4.74)}$ | $\mathbf{46.60}_{(11.82)}$ | $\mathbf{0.60}_{(0.12)}$ | $0.57_{(0.10)}$ | $\mathbf{0.63}_{(0.17)}$ |
| | SCORE | Base | $13.00_{(3.92)}$ | $32.60_{(12.94)}$ | $0.68_{(0.12)}$ | $\mathbf{0.71}_{(0.11)}$ | $0.66_{(0.15)}$ |
| | | PEP | $\mathbf{11.60}_{(3.36)}$ | $\mathbf{21.00}_{(14.51)}$ | $\mathbf{0.75}_{(0.09)}$ | $0.70_{(0.09)}$ | $\mathbf{0.81}_{(0.16)}$ |
| | NoGAM | Base | $13.80_{(3.26)}$ | $37.00_{(12.44)}$ | $0.65_{(0.09)}$ | $0.69_{(0.09)}$ | $0.63_{(0.11)}$ |
| | | PEP | $\mathbf{11.40}_{(3.65)}$ | $\mathbf{27.60}_{(19.55)}$ | $\mathbf{0.72}_{(0.11)}$ | $0.69_{(0.09)}$ | $\mathbf{0.78}_{(0.17)}$ |
| | DiffAN | Base | $18.70_{(5.23)}$ | $51.60_{(10.71)}$ | $0.54_{(0.13)}$ | $0.52_{(0.14)}$ | $0.57_{(0.13)}$ |
| | | PEP | $\mathbf{15.20}_{(4.15)}$ | $\mathbf{47.60}_{(13.22)}$ | $\mathbf{0.60}_{(0.13)}$ | $\mathbf{0.58}_{(0.10)}$ | $\mathbf{0.63}_{(0.19)}$ |
| | CaPS | Base | $15.00_{(3.03)}$ | $34.33_{(6.86)}$ | $0.65_{(0.07)}$ | $0.63_{(0.07)}$ | $0.68_{(0.08)}$ |
| | | PEP | $\mathbf{8.75}_{(3.95)}$ | $\mathbf{26.25}_{(14.52)}$ | $\mathbf{0.79}_{(0.10)}$ | $\mathbf{0.87}_{(0.11)}$ | $\mathbf{0.73}_{(0.12)}$ |
| Measure-Err | CAM | Base | $\mathbf{19.20}_{(1.75)}$ | $66.60_{(10.28)}$ | $0.33_{(0.09)}$ | $\mathbf{0.49}_{(0.14)}$ | $0.25_{(0.07)}$ |
| | | PEP | $19.60_{(1.14)}$ | $\mathbf{61.00}_{(9.90)}$ | $\mathbf{0.34}_{(0.07)}$ | $0.46_{(0.09)}$ | $\mathbf{0.27}_{(0.06)}$ |
| | SCORE | Base | $15.70_{(2.71)}$ | $49.10_{(10.33)}$ | $0.54_{(0.10)}$ | $\mathbf{0.78}_{(0.15)}$ | $0.42_{(0.11)}$ |
| | | PEP | $\mathbf{14.60}_{(2.51)}$ | $\mathbf{40.60}_{(11.44)}$ | $\mathbf{0.58}_{(0.09)}$ | $0.76_{(0.09)}$ | $\mathbf{0.48}_{(0.12)}$ |
| | NoGAM | Base | $15.30_{(2.58)}$ | $47.90_{(13.36)}$ | $0.55_{(0.09)}$ | $\mathbf{0.81}_{(0.15)}$ | $0.42_{(0.10)}$ |
| | | PEP | $\mathbf{14.00}_{(3.54)}$ | $\mathbf{36.20}_{(15.66)}$ | $\mathbf{0.60}_{(0.14)}$ | $0.79_{(0.13)}$ | $\mathbf{0.49}_{(0.15)}$ |
| | DiffAN | Base | $\mathbf{18.14}_{(2.79)}$ | $57.71_{(9.74)}$ | $\mathbf{0.45}_{(0.08)}$ | $\mathbf{0.57}_{(0.14)}$ | $\mathbf{0.38}_{(0.07)}$ |
| | | PEP | $19.00_{(2.16)}$ | $\mathbf{57.25}_{(13.89)}$ | $0.42_{(0.08)}$ | $0.52_{(0.13)}$ | $0.35_{(0.05)}$ |
| | CaPS | Base | $\mathbf{15.83}_{(1.47)}$ | $\mathbf{43.83}_{(9.83)}$ | $\mathbf{0.55}_{(0.01)}$ | $0.68_{(0.09)}$ | $\mathbf{0.47}_{(0.04)}$ |
| | | PEP | $16.00_{(0.82)}$ | $46.25_{(7.54)}$ | $0.51_{(0.04)}$ | $\mathbf{0.82}_{(0.07)}$ | $0.38_{(0.03)}$ |
| Non-i.i.d | CAM | Base | $15.20_{(4.66)}$ | $53.20_{(18.34)}$ | $0.55_{(0.15)}$ | $\mathbf{0.64}_{(0.20)}$ | $0.49_{(0.12)}$ |
| | | PEP | $\mathbf{13.60}_{(6.07)}$ | $\mathbf{32.80}_{(14.11)}$ | $\mathbf{0.66}_{(0.17)}$ | $0.64_{(0.18)}$ | $\mathbf{0.69}_{(0.17)}$ |
| | SCORE | Base | $14.80_{(4.29)}$ | $54.60_{(21.80)}$ | $0.56_{(0.14)}$ | $0.66_{(0.15)}$ | $0.50_{(0.16)}$ |
| | | PEP | $\mathbf{11.80}_{(4.97)}$ | $\mathbf{34.80}_{(21.12)}$ | $\mathbf{0.71}_{(0.12)}$ | $\mathbf{0.68}_{(0.14)}$ | $\mathbf{0.74}_{(0.13)}$ |
| | NoGAM | Base | $15.30_{(4.30)}$ | $54.40_{(21.16)}$ | $0.55_{(0.14)}$ | $0.64_{(0.16)}$ | $0.49_{(0.14)}$ |
| | | PEP | $\mathbf{10.40}_{(4.67)}$ | $\mathbf{34.80}_{(24.16)}$ | $\mathbf{0.73}_{(0.12)}$ | $\mathbf{0.69}_{(0.11)}$ | $\mathbf{0.78}_{(0.15)}$ |
| | DiffAN | Base | $17.60_{(5.76)}$ | $56.50_{(17.75)}$ | $0.51_{(0.15)}$ | $0.54_{(0.18)}$ | $0.48_{(0.13)}$ |
| | | PEP | $\mathbf{14.80}_{(4.32)}$ | $\mathbf{46.00}_{(11.77)}$ | $\mathbf{0.62}_{(0.12)}$ | $\mathbf{0.60}_{(0.11)}$ | $\mathbf{0.66}_{(0.15)}$ |
| | CaPS | Base | $14.50_{(4.50)}$ | $48.90_{(20.79)}$ | $0.60_{(0.15)}$ | $0.60_{(0.11)}$ | $0.61_{(0.05)}$ |
| | | PEP | $\mathbf{11.83}_{(2.32)}$ | $\mathbf{42.33}_{(12.64)}$ | $\mathbf{0.68}_{(0.04)}$ | $\mathbf{0.77}_{(0.09)}$ | $0.61_{(0.05)}$ |
| Unfaithful | CAM | Base | $11.90_{(3.84)}$ | $45.70_{(12.22)}$ | $0.63_{(0.11)}$ | $\mathbf{0.67}_{(0.13)}$ | $0.61_{(0.10)}$ |
| | | PEP | $\mathbf{11.50}_{(1.91)}$ | $\mathbf{30.00}_{(7.62)}$ | $\mathbf{0.71}_{(0.04)}$ | $0.65_{(0.05)}$ | $\mathbf{0.77}_{(0.02)}$ |
| | SCORE | Base | $6.50_{(0.93)}$ | $24.75_{(8.31)}$ | $0.83_{(0.03)}$ | $\mathbf{0.87}_{(0.05)}$ | $0.80_{(0.04)}$ |
| | | PEP | $\mathbf{5.80}_{(1.64)}$ | $\mathbf{10.20}_{(9.20)}$ | $\mathbf{0.87}_{(0.04)}$ | $0.81_{(0.05)}$ | $\mathbf{0.95}_{(0.03)}$ |
| | NoGAM | Base | $6.10_{(2.08)}$ | $25.40_{(12.59)}$ | $0.84_{(0.07)}$ | $\mathbf{0.89}_{(0.10)}$ | $0.19_{(0.01)}$ |
| | | PEP | $\mathbf{5.80}_{(3.03)}$ | $\mathbf{14.60}_{(17.97)}$ | $\mathbf{0.86}_{(0.09)}$ | $0.80_{(0.10)}$ | $\mathbf{0.93}_{(0.09)}$ |
| | DiffAN | Base | $12.75_{(1.67)}$ | $46.62_{(12.16)}$ | $0.64_{(0.06)}$ | $\mathbf{0.64}_{(0.05)}$ | $0.66_{(0.08)}$ |
| | | PEP | $\mathbf{11.25}_{(1.71)}$ | $\mathbf{38.75}_{(19.35)}$ | $\mathbf{0.69}_{(0.08)}$ | $0.64_{(0.05)}$ | $\mathbf{0.77}_{(0.12)}$ |
| | CaPS | Base | $5.10_{(1.52)}$ | $17.20_{(5.20)}$ | $0.87_{(0.04)}$ | $0.90_{(0.04)}$ | $0.92_{(0.05)}$ |
| | | PEP | $\mathbf{3.78}_{(1.86)}$ | $\mathbf{12.11}_{(7.13)}$ | $\mathbf{0.91}_{(0.04)}$ | $\mathbf{0.91}_{(0.05)}$ | $0.92_{(0.05)}$ |

