# OpenReview forum: "Prequential Evidence Pruning: Information-Theoretic Edge Selection for Ordering-Based Causal Discovery"
_ICLR.cc/2026/Conference — Submitted to ICLR 2026_

### Official Review · Reviewer_E4Pd · 2025-10-15

**Soundness:** 2
**Presentation:** 2
**Contribution:** 2
**Rating:** 4
**Confidence:** 4

**Summary:**

The paper proposes Prequential Evidence Pruning (PEP), a framework to improve ordering-based causal discovery. It replaces traditional pruning heuristics with a principled cost-benefit analysis, where the out-of-sample predictive evidence for an edge is weighed against a computable MDL complexity penalty. The method is shown to be a highly effective "plug-and-play" module that consistently improves the performance of ordering-based algorithms on small-scale data.

**Strengths:**

The paper’s key strength lies in its reframing of the pruning problem. The information-theoretic foundation connects the evidence metric to Conditional Mutual Information, and the resulting method addresses the known brittleness of marginal, assumption-heavy pruning techniques

**Weaknesses:**

The paper's impact is severely limited by its failure to address the critical challenge of scalability, which is a primary focus of the causal discovery community.
1. The primary weakness is that this work provides an enhancement to a paradigm (ordering-based search) that is fundamentally constrained by exponential time complexity. The current research frontier in causal discovery is focused on overcoming this exact limitation through scalable, gradient-based methods that reframe the problem for continuous optimization. By focusing on a search-based paradigm, this work feels out of step with the direction the field is heading to solve large-scale problems.
2.  It is a local improvement that does not resolve the global bottleneck of the search-based approach. The paper's own analysis confirms a pruning cost that is quadratic in the number of candidate parents, and the exponential complexity of the broader search remains the limiting factor. This restricts the method's practical applicability to the large-scale datasets where new causal discovery methods are most critically needed.
3. The experiments are confined to small graphs with 10 to 20 nodes. A significant contribution in causal discovery must demonstrate its relevance to larger, more challenging problems. There is no evidence or discussion of how this method would perform on graphs with 100 or 200 nodes. Without a comparison showing that a PEP-enhanced method can compete with leading gradient-based methods in terms of accuracy and time efficiency at a larger scale, the paper's claims of broad utility are unsubstantiated. It perfects a method within a specific niche without challenging the very paradigms developed to overcome that niche's limitations.

**Questions:**

1. Figure 5 shows that PEP's significant performance gains are realized almost exclusively when using the powerful, pre-trained TabPFN model, while its performance with standard learners like XGBoost is far less compelling. How can you disentangle the contribution of the PEP framework itself from the exceptional zero-shot performance of its predictive engine? Does this reliance on a large foundation model limit the practical accessibility of your method?
2. The experiments demonstrate strong performance on graphs with up to 20 nodes. However, the field is increasingly focused on scalable, gradient-based methods for larger problems. Given that PEP operates within the computationally expensive ordering-based paradigm, how do you justify its relevance for real-world applications where the number of variables is often in the hundreds or thousands? Can you provide results on larger scale data and real-world data?
3. Given the computational constraints, what do you see as the primary use case for PEP? Is it best suited for small, high-stakes problems where principled, auditable decisions are critical, or do you envision a path to making it viable for larger-scale exploratory causal analysis?

---

> ### Author Response · Authors · 2025-12-03
>
> We thank the reviewer for the critical assessment regarding scalability. We agree that scalability is a paramount concern. Below, we address your concerns by re-evaluating the paradigm and detailing our specific scalability improvements.
>
> ---
>
> > **[W1]** *The primary weakness is that this work provides an enhancement to a paradigm (ordering-based search) that is fundamentally constrained by exponential time complexity... By focusing on a search-based paradigm, this work feels out of step with the direction the field is heading to solve large-scale problems.*
>
> We agree that continuous optimization methods can be very attractive for truly large scale settings, especially when the number of variables is in the hundreds or thousands. Our choice to work within an ordering based paradigm is not meant to dispute this, but rather to prioritize structural validity and stability in the regime we target.
>
> 1. **Structural validity (acyclicity by construction)**
>     * Continuous approaches such as NOTEARS [1] or DAGMA [2] must enforce a global acyclicity constraint, typically encoded as a function $h(W) = 0$ in the weight matrix. In practice this constraint is only satisfied approximately, and it is common to obtain graphs that require additional post processing to remove small cycles.
>     * In contrast, ordering based methods first obtain a topological order and then learn parents that respect this order, which guarantees acyclicity by construction. For high stakes applications where producing a valid DAG is critical, this structural guarantee is an important advantage.
>
> 2. **Optimization stability**
>     * The objectives used in continuous formulations are highly non convex, which makes them sensitive to initialization and prone to getting trapped in local optima. Their performance can also depend on data specific artifacts such as variance scaling.
>     * By conditioning on an order and reducing structure learning to a set of supervised parent selection problems, ordering based pipelines avoid these global non convex acyclicity constraints. In our experiments this leads to more stable and consistent behavior across mechanisms and misspecification scenarios.
>
> Within this paradigm, our contribution is to strengthen the pruning stage with a prequential, MDL based rule and a hierarchical group pruning strategy, which together make ordering based pipelines more practical in the sparse, moderate to large dimensional regime we study. We do not claim to replace continuous methods at extreme scales, but we believe that ordering based approaches with guaranteed acyclicity remain a relevant and complementary direction for reliable causal discovery.
>
> ---
>
> **Reference**
>
> [1] Zheng et al., "Dags with no tears: Continuous optimization for structure learning," NeurIPS, 2018.
>
> [2] Bello et al., "Dagma: Learning dags via m-matrices and a log-determinant acyclicity characterization," NeurIPS, 2022.

---

> ### Author Response · Authors · 2025-12-03
>
> > **[W2]** *It is a local improvement that does not resolve the global bottleneck... The paper's own analysis confirms a pruning cost that is quadratic... This implies the pruning cost is at least $O(d^2)$... restricts the method's practical applicability.*
>
> This was a valid critique of the initial submission. In the revision, we introduced a *Hierarchical Group Pruning* strategy that substantially reduces the pruning cost.
>
> * **Divide-and-conquer.** Instead of testing parents one by one, we recursively test groups of parents and only split groups whose joint evidence is close to or above the MDL gate.
> * **Complexity reduction.** As shown in **Appendix B.4**, under a sparse regime with at most $s$ true parents per node, this reduces the number of evidence evaluations per node from $\Theta(P_j^2)$ to $O(s \log P_j)$, where $P_j$ is the number of admissible parents. This directly alleviates the quadratic pruning bottleneck of the original scheme.
> * **Runtime.** We added **Figure 6**, which shows that although PEP has noticeable overhead at small $d$, its runtime grows more slowly than the baseline as $d$ increases. At $d = 100$, PEP is significantly faster than the standard CAM pruning (about 7.5x speedup in our setup).
>
> **[Action]** We added **Section 3.4** (Hierarchical Group Pruning) and **Figure 6** (Runtime scalability) to show that, in the sparse regimes we target, the quadratic pruning bottleneck is substantially mitigated in both theory and practice.
>
> ---
>
> > **[W3]** *The experiments are confined to small graphs with 10 to 20 nodes... There is no evidence or discussion of how this method would perform on graphs with 100 or 200 nodes.*
>
> We have significantly expanded our experimental suite to cover larger graphs.
>
> * **Large-scale experiments.** We added **Table 4**, which reports results for graphs with $d = 30, 50,$ and $100$ nodes across multiple mechanisms and backbones.
> * **Results.** PEP consistently improves over the baseline pruning across all reported dimensions. In several cases, the performance gap *widens* as $d$ increases (for example, on SCORE at $d = 100$, PEP improves F1 by about 14%), indicating that the adaptive MDL gate continues to work well as the search space grows and does not break down in larger settings.
>
> **[Action]** We added **Table 4** and updated **Figure 5** to show consistent performance gains up to $d = 100$, together with a discussion in the text about how PEP behaves as the dimensionality increases.

---

> ### Author Response · Authors · 2025-12-03
>
> > **[Q1]** *Figure 5 shows that PEP's... performance gains are realized almost exclusively when using... TabPFN... How can you disentangle the contribution of the PEP framework itself from the exceptional zero-shot performance of its predictive engine?*
>
> We disentangle this by testing PEP with a diverse set of standard predictors (Random Forest, XGBoost, CatBoost, LightGBM, MITRA), using Platt scaling to improve calibration.
>
> * **Results.** As shown in the new **Table 3**, PEP consistently improves over the CAM pruning baseline across these predictors, and is never substantially worse. The gains are typically smaller than with TabPFN, which is expected since TabPFN is more strongly calibrated, but the pattern is robust across models.
> * **Interpretation.** This suggests that the benefit comes from the PEP framework itself, that is, context aware prequential evidence combined with the MDL gate, rather than solely from TabPFN. TabPFN should be viewed as a particularly strong instantiation of this framework, not as a requirement.
>
> **[Action]** We added **Table 3**, which reports a sweep over multiple predictors with and without PEP, and discuss these results in the revised experimental section.
>
> ---
>
> > **[Q2]** *Given that PEP operates within the computationally expensive ordering-based paradigm, how do you justify its relevance for real-world applications where the number of variables is often in the hundreds or thousands?*
>
> We justify relevance in terms of *precision*, *flexibility*, and *improved efficiency* in the regime we currently target.
>
> * **Precision.** In high stakes domains (for example healthcare or biology), false positives are costly. In our experiments, ordering based pipelines augmented with PEP achieve lower SHD and improved precision compared to their original pruning rules and to several continuous optimization baselines, especially in non linear and misspecified settings. This makes PEP attractive when correctness is more important than raw speed.
>
> * **Flexibility of the predictive backbone.** PEP is a plug in pruning rule that only requires log likelihoods from a predictive component. For large datasets or higher dimensional problems where TabPFN is too expensive, users can swap in faster models (for example gradient boosting, as in **Table 3**) and still benefit from the same PEP decision rule.
>
> * **Improved efficiency via hierarchical pruning.** In the revision, we introduce Hierarchical Group Pruning (**Section 3.4**), which reduces the number of evidence evaluations from a naive quadratic dependence on the number of candidate parents to $O(s \log P_j)$ per node under sparsity (at most $s$ true parents), as detailed in **Appendix B.4**. In our runtime experiments up to $d = 100$ variables (**Figure 6**), this makes the pruning step substantially cheaper and no longer the dominant bottleneck in the tested regime.
>
> **[Action]** We added **Section 3.4** (hierarchical pruning and its complexity sketch) and updated **Section 4** with runtime and large graph experiments (up to $d \approx 100$) to document the regime in which PEP is currently practical, while outlining scaling beyond this as future work.

---

> ### Author Response · Authors · 2025-12-04
>
> > **[Q3]** *Is it best suited for small, high-stakes problems... or do you envision a path to making it viable for larger-scale exploratory causal analysis?*
>
> We see PEP as a **general-purpose pruning framework** whose current strengths are (i) robust inference in “misspecify” real-world scenarios and (ii) increasingly practical scalability for moderate-to-large graphs.
>
> 1. **Robustness for small–medium, high-stakes problems.** PEP is particularly attractive when the data-generating process is only partially known or clearly misspecified.
>
>    * **Resilience to misspecification.** In our stress-test suite (**Figure 3**), PEP maintains strong structural accuracy under confounding, measurement error, and non-i.i.d. sampling, whereas several baselines degrade markedly.
>    * **Functional agnosticism.** Across mechanisms ranging from fully linear to fully non-linear (**Figure 4**), PEP remains competitive without assuming a specific functional form, making it a safe default when model misspecification is a concern.
>
> 2. **Path towards larger-scale exploratory analysis.** In the revision, we have made PEP substantially more scalable, so that it is also viable for exploratory analyses on larger graphs.
>
>    * **Algorithmic side.** The new *Hierarchical Group Pruning* strategy (**Section 3.4, Appendix B.4**) reduces the number of evidence evaluations from a naive quadratic dependence in the number of candidate parents to $O(s \log P_j)$ per node under sparsity (at most $s$ true parents). This alleviates the classical quadratic pruning bottleneck of ordering-based methods without changing the underlying decision rule.
>    * **Empirical side.** Our new experiments on graphs with up to $d = 100$ variables (**Figure 6, Table 4**) show that PEP can handle such structures within minutes and achieves a substantial speedup (around $7.5\times$ faster than the baseline in our setting), while preserving or improving structural accuracy.
>
> **[Action]** We highlight robustness under various real-data and misspecified scenarios in **Section 4.1**. We document the improved scalability and large-graph behavior in **Section 4.2 and Figure 6 / Table 4**, explicitly stating the regime (sparse graphs up to $d \approx 100$) in which PEP is currently practical.
>
> Overall, our view is that PEP is already well suited for small- to medium-scale, high-stakes problems that demand robustness, and the new hierarchical pruning strategy moves it closer to being a practical tool for larger-scale exploratory causal discovery as well.

---

### Official Review · Reviewer_zpst · 2025-10-18

**Soundness:** 2
**Presentation:** 1
**Contribution:** 2
**Rating:** 2
**Confidence:** 3

**Summary:**

The authors introduce Prequential Evidence Pruning (PEP) as a plug-and-play technique to improve existing methods for topological ordering-based causal discovery. Briefly, PEP takes a topological order of a causal graph and iteratively refines it by pruning edges that provide an information gain smaller than a Minimum Description Length (MDL)-based threshold. Experiments illustrate that incorporating PEP into other methods leads to significant improvements with respect to standard metrics for causal discovery (SHD, SID, and F1).

Although the work provides an interesting perspective on a specific class of causal discovery algorithms, it employs a significantly heavy language that hinders my attempts to clearly understand the proposed method.  For example, it states that PEP captures “synergistic and non-additive interactions”, and that it differs from other methods by an “evidence semantics” through a score that “concentrates under cross-fitting” - with the meaning of these expressions remaining elusive throughout the manuscript.

**Strengths:**

1. Algorithm 1 provides a clear description of PEP.

2. Experimental results highlight PEP’s effectiveness.

**Weaknesses:**

My main concern with the work regards the excessive use of hand-waved expressions and formulas that are hard to understand. On top of the ones presented in the Summary section, I also have the following remarks.

1. Expectation operators do not specify with respect to which distribution the expectations are being computed.

2. Corollary 2 uses the term $P_{j}$, which is only introduced later in the text. This is also true for $\text{Pred}_{\pi}$.

3. Also, it is unclear what the authors mean by “marginal additivity-constrained methods" - and how PEP can circumvent this presumed issue.

4. It is correspondingly confusing what the authors mean by “the price of order-aware combinatorics”. Could the authors elaborate on this?

5. Proposition 1 is used to support the claim that “the statistic remains well-behaved with imperfect predictors”. However, both the meanings of “statistic” (perhaps the authors are referring to $\delta$?) and the fact that it is well-behaved under imperfect predictors (which predictors?) are not clearly represented. (This is also connected to (1); expectation operators are not clearly described).

6. Corollary 1 talks about small regrets; how small, and how near is $\delta$ to $0$?

7. Figure 4 is also difficult to parse: how is each axis measured and what is the baseline for the $\Delta \text{Area}$ calculations?

8. In Figure 7, how is “linearity” measured?

**Questions:**

Please refer to the questions above.

---

> ### Author Response · Authors · 2025-12-03
>
> We thank the reviewer for their careful reading and for highlighting issues regarding clarity, notation, and definitions. We have substantially revised the manuscript to improve mathematical precision and readability. Below, we address each of your concerns point by point, referencing the specific improvements in the **revised paper**.
>
> ---
>
> > **[Summary]** *Although the work provides an interesting perspective... it employs a significantly heavy language that hinders my attempts to clearly understand the proposed method. For example... "synergistic and non-additive interactions", ... "evidence semantics", ... "concentrates under cross-fitting".*
>
> We appreciate this feedback and agree that the initial wording was not sufficiently precise. In the revision, we have refined these terms and provided concrete definitions:
>
> * **Synergistic / non-additive interactions.** We now explicitly define this notion in the **Introduction** and provide a dedicated **Appendix D**. In particular, **Section D.1** presents the noisy XOR example, where the marginal signal is zero ($I(X_j; X_i) \approx 0$) but the conditional signal is strictly positive ($I(X_j; X_i \mid X_k) > 0$). This example makes precise what we mean by synergy and how PEP can capture such non-additive interactions.
>
> * **Evidence semantics.** By this we mean that the edge score is interpreted as a log-evidence (log-likelihood) gain rather than as a $p$-value. In **Section 3.1**, we define the prequential log-evidence gain $\delta_{i \to j}(q; S)$ as a per-sample log-likelihood ratio between models with and without the candidate parent, and in **Section 3.2** we show that, under an oracle predictor, its population target equals the conditional mutual information (CMI). This connects the statistic to a standard information-theoretic quantity and clarifies the intended semantics.
>
> * **Concentrates under cross-fitting.** We have formalized this statement in **Theorem 2 (Section 3.2)**, which proves that the prequential evidence statistic $\delta_{i \to j}(q; S)$ concentrates around its mean with high probability under a sub-exponential tail condition. The proof explicitly uses the independence of held-out folds induced by our sample-splitting (cross-fitting) scheme.
>
> **[Action]** Added explicit definitions and explanations in the **Introduction**, **Section 3.1**, and **Section 3.2**, and included concrete mathematical examples in **Appendix D** to clarify these terms.
>
> ---
>
> > **[W]** *My main concern with the work regards the excessive use of hand-waved expressions and formulas...*
>
> We have thoroughly revised **Section 3** to replace heuristic or informal descriptions with formal definitions, precise assumptions, and theorem statements. We also added a **notation table (Table B.1 in Appendix B)** that collects all symbols used in the paper and points to their definitions, so that the mathematical presentation can be followed unambiguously.
>
> **[Action]** Rewrote **Section 3** to improve mathematical rigor and added **Table B.1** in **Appendix B** as a central reference for all notation.

---

> ### Author Response · Authors · 2025-12-03
>
> > **[W1]** *Expectation operators do not specify with respect to which distribution the expectations are being computed.*
>
> Thank you for pointing this out. In **Section 3** of the revision, we now explicitly state that *$\mathbb{E}$ denotes expectation with respect to the true data generating distribution $p$ unless stated otherwise*. When expectations are taken with respect to another distribution, we write it explicitly (for example, $\mathbb{E}_q$). This convention is applied consistently throughout the theoretical results in **Section 3.2** and the proofs in **Appendix E**.
>
> **[Action]** Added a formal definition of the expectation operator $\mathbb{E}$ in **Section 3** and use explicit subscripts whenever expectations are taken under distributions other than $p$.
>
> ---
>
> > **[W2]** *Corollary 2 uses the term $P_j$, which is only introduced later in the text. This is also true for $Pred_\pi$.*
>
> We have corrected the order of definitions. The predecessor set $Pred_\pi(j)$ of node $j$ under the topological order $\pi$ and its cardinality $P_j = |Pred_\pi(j)|$ are now defined early in **Section 3**, before they are used in **Corollary 2** and in the MDL gate formulation in **Eq. (3)**.
>
> **[Action]** Reorganized **Section 3** so that $Pred_\pi(j)$ and $P_j$ are introduced before their first appearance in theorems, corollaries, and MDL expressions.
>
> ---
>
> > **[W3]** *Also, it is unclear what the authors mean by "marginal additivity-constrained methods" and how PEP can circumvent this presumed issue.*
>
> We have clarified this distinction in the **Introduction** and **Section 2**.
>
> - **Meaning.** Standard methods such as CAM pruning evaluate each parent candidate $X_i$ individually (marginally) using a generalized additive model (GAM), testing the null hypothesis $f_i(X_i) = 0$. This implicitly constrains the total effect to be a sum of individual additive effects.
> - **PEP’s solution.** As illustrated in **Figure 1** and formally defined in **Eq. (1)**, PEP evaluates the *conditional* gain of adding a parent $X_i$ given the current set of co-parents $X_{S \setminus \{i\}}$. By measuring $\log q(x_j \mid x_S) - \log q(x_j \mid x_{S \setminus \{i\}})$, PEP captures interactions (synergies) that are invisible to marginal additive tests.
>
> **[Action]** Expanded **Section 2** to contrast marginal testing with conditional evidence and updated **Figure 1** to visually highlight how PEP detects synergistic effects.
>
> ---
>
> > **[W4]** *It is correspondingly confusing what the authors mean by "the price of order-aware combinatorics". Could the authors elaborate on this?*
>
> We have rephrased this for clarity in **Section 3.3** and **Appendix B.1**. The phrase refers to the specific code length required to describe the selection of a parent under the constraint of a fixed topological order.
>
> - Instead of a generic penalty, **Eq. (3)** now explicitly writes this cost as $\ln(P_j - k)$, which is the information required to identify which parent is added from the $P_j - k$ remaining valid candidates for node $j$ at step $k$. This provides a precise information theoretic justification for the penalty term.
>
> **[Action]** Refined the text in **Section 3.3** and added a detailed derivation of this combinatorial cost in **Appendix B.1**.

---

> ### Author Response · Authors · 2025-12-03
>
> > **[W5]** *Proposition 1 is used to support the claim that "the statistic remains well-behaved with imperfect predictors". However, both the meanings of "statistic" and "well-behaved" are not clearly represented.*
>
> We have rewritten **Proposition 1 in Section 3.2** to be more precise.
>
> - **The statistic.** It is the prequential log evidence gain $\delta_{i \to j}(q; S)$ defined in **Eq. (1)**.
> - **Well behaved.** Proposition 1 now states that the difference between the estimated evidence $\delta_{i \to j}(q; S)$ and the population conditional mutual information is bounded by the sum of the log loss regrets $r_S$ and $r_{S \setminus \{i\}}$ of the predictor. This shows that as long as the predictor has small regret, the pruning statistic remains close to its population target and the resulting pruning decision is reliable.
>
> **[Action]** Revised **Proposition 1** to explicitly name the statistic and to bound its bias in terms of log loss regret.
>
> ---
>
> > **[W6]** *Corollary 1 talks about small regrets; how small, and how near is $\delta$ to 0?*
>
> We have clarified the phrasing in **Corollary 1**.
>
> - By “small regret” we mean that the predictive distribution $q$ is close to the true distribution $p$ in KL divergence, so that $r_S$ and $r_{S \setminus \{i\}}$ are small.
> - Under conditional independence (i.e., when $X_i$ is not a parent of $X_j$), the population conditional mutual information is exactly $0$. The corollary states that in this case the empirical statistic $\delta_{i \to j}(q; S)$ is upper bounded by the sum of the regret terms plus a finite sample fluctuation term controlled by **Theorem 2**, which decays exponentially with the sample size. This ensures that non parent edges have evidence close to $0$ and are effectively pruned.
>
> **[Action]** Clarified **Corollary 1** to relate “small regret” to closeness in KL divergence and to make the bound on $\delta_{i \to j}(q; S)$ explicit.
>
> ---
>
> > **[W7]** *Figure 4 is also difficult to parse: how is each axis measured and what is the baseline for the $\Delta$Area calculations?*
>
> We have updated the figure (now **Figure 3** in the revision) and its caption to be self contained.
>
> - **Axes.** The axes represent **normalized inverted SHD** (i.e., (1 - SHD) /max_edges), **normalized inverted SID**, and **F1 score**, with details given in **Appendix F.3**. For all axes, $1.0$ (outermost) corresponds to the best performance and $0.0$ (center) to the worst.
> - **Baseline.** The $\Delta$Area reported in the legend is the percentage increase in polygon area achieved by the PEP augmented method (colored line) relative to its baseline pruning method (gray line). A positive $\Delta$Area indicates that PEP improves the method jointly across all three metrics.
>
> **[Action]** Updated **Figure 3** and its caption and added **Appendix F.3** to precisely define the axis normalizations and the $\Delta$Area computation.
>
> ---
>
> > **[W8]** *In Figure 7, how is "linearity" measured?*
>
> We have clarified the experimental setup in **Section 4** (“Robustness to Functional Form”) and **Appendix F.1**.
>
> - The x axis represents the **linear proportion** $\rho_{\text{lin}}$ used in the data generating process.
> - For each node, its structural function is chosen to be linear with probability $\rho_{\text{lin}}$ and nonlinear (for example, a neural network or Gaussian process) with probability $1 - \rho_{\text{lin}}$.
> - $\rho_{\text{lin}} = 0.0$ corresponds to fully nonlinear mechanisms and $\rho_{\text{lin}} = 1.0$ to fully linear mechanisms. The results (now shown in **Figure 4**) demonstrate that PEP is robust across this entire spectrum.
>
> **[Action]** Added a detailed description of the linear proportion parameter $\rho_{\text{lin}}$ in **Section 4** and **Appendix F.1**, and updated the figure caption accordingly.

---

### Official Review · Reviewer_Mmos · 2025-10-23

**Soundness:** 2
**Presentation:** 3
**Contribution:** 3
**Rating:** 4
**Confidence:** 3

**Summary:**

This paper addresses a key bottleneck in "Ordering-based Causal Discovery" methods—the pruning stage—by proposing a novel, information-theoretic framework called "Prequential Evidence Pruning" (PEP). The authors point out that existing pruning methods (like CAM pruning) heavily rely on marginal tests, additivity assumptions, and manually-tuned thresholds. This causes them to fail in capturing non-additive interactions (such as synergies) and compromises reproducibility.The PEP framework reframes the pruning problem as a local, information-theoretic cost-benefit analysis. For each candidate edge $i \rightarrow j$, the "benefit" is quantified by a "Prequential Log-Evidence Gain," which is the improvement in the prequential (i.e., out-of-fold) predictive log-likelihood for the child $X_j$ when conditioning on $X_i$ in addition to its co-parents $S \setminus \{i\}$. The population target of this gain is equal to the Conditional Mutual Information (CMI). The "cost" is an adaptive code-length penalty computed according to the Minimum Description Length (MDL) principle, which adjusts to sample size, the number of admissible parents, and the current parent set size. An edge is retained only when this benefit (evidence gain) exceeds the cost (MDL penalty).

**Strengths:**

The paper accurately identifies a core pain point in ordering-based causal discovery methods: the pruning stage. The comparison in Figure 1 between the "Keyhole view" (marginal) and the "Panoramic view" (context-aware) very intuitively illustrates the limitations of existing methods.

The misspecification stress test in Figure 4 is very comprehensive. PEP's advantages are clearly validated, especially in the Post-Nonlinear (PNL) setting and in the robustness to functional form test (Figure 7), which directly confirms the paper's hypothesis.

The experiment using a random topological order in Table 2 effectively isolates the performance of the pruning stage. This demonstrates that PEP's advantage is not just "piggybacking" on a strong ordering algorithm, but stems from the superiority of its own local decision rule.

**Weaknesses:**

Although the paper presents the use of TabPFN as an advantage (zero-shot, well-calibrated), this is also its main weakness. The results in Figure 5 show that when PEP is paired with RF or XGBoost, its performance has no significant advantage over CAM pruning, and is even worse in some cases. This strongly suggests that the practical performance of the PEP framework is highly dependent on a powerful and well-calibrated density estimator like TabPFN. This weakens the paper's claim of being "model-class agnostic". In domains where TabPFN is not applicable or performs poorly (e.g., high-dimensional data, non-tabular data), the effectiveness of the PEP framework would be highly questionable.

All synthetic data experiments are limited to $d=10$ nodes (Sachs also has only $d=11$). This is a very small scale. Algorithm 1 implements greedy backward pruning. The computational analysis in Appendix H.1 shows that PEP's cost is on the order of $O(Knm_j^2\overline{\alpha})$, where $m_j$ is the number of candidate parents. This is in the same complexity class as CAM pruning's $O(B s^2 n m_j^2)$ (both are quadratic in $m_j$). When the graph density increases, or under the stress test of a random order (as in Table 2), $m_j$ could approach $O(d)$. This implies the pruning cost is at least $O(d^2)$ (per node), leading to a total complexity of $O(d^3)$ or even $O(d^4)$. The paper only reports runtime for $d=10$ (Table H.1), where its runtime is already noticeably higher than CAM pruning.

A core argument of the paper is replacing "manually-tuned thresholds". However, its MDL gate (Eq. 3) includes a fixed overhead $\kappa$. In Appendix G, the authors state that $\lambda=1$ and $\kappa=25$ is "calibrated once for the entire study". This feels like just swapping one "manually-tuned $\alpha$" for another "manually-tuned $\kappa$". Although the sensitivity analysis in Figure 6 shows that the computed MDL gate (marked with $\star$) lies within a flat, high-performance plateau, this analysis does not explore how the position of this $\star$ mark, and consequently the final SHD/SID, would be affected if $\kappa$ took different values.

**Questions:**

Given the quadratic dependency on $m_j$ (number of candidate parents) in the complexity analysis and the limitation of experiments to $d=10$, could the authors provide an experiment on scaling with the number of nodes $d$?

The results in Figure 5 show that RF/XGB perform poorly , and the authors speculate this is because TabPFN provides "high-fidelity calibrated densities". This speculation is very reasonable. It is suggested that the authors add an experiment: when using XGBoost or RF as the predictor, add an extra "post-hoc calibration" step (e.g., Isotonic Regression or Platt Scaling) before feeding the outputs into the PEP framework.

Algorithm 1 uses Greedy Backward Elimination. What is the reason for choosing this strategy? At the beginning of backward elimination, the context $S$ used to evaluate $\delta_{i\rightarrow j}$ contains many irrelevant variables, which could be computationally expensive and interfere with the detection of true synergies.

---

> ### Author Response · Authors · 2025-12-03
>
> We appreciate the reviewer’s insightful critique regarding scalability, model dependence, and the nature of the MDL gate. These comments have driven us to implement significant upgrades to the framework, including a new hierarchical pruning algorithm and a rigorous re-formulation of the penalty term. Below, we address your concerns point-by-point.
>
> ---
>
> > **[W1]** *The results in Figure 5 show that when PEP is paired with RF or XGBoost, its performance has no significant advantage over CAM pruning... This strongly suggests that the practical performance of the PEP framework is highly dependent on a powerful and well-calibrated density estimator like TabPFN.*
>
> We agree that the quality of the evidence score depends on the calibration of the predictor. Our aim, however, is to show that the *framework* is not tied to TabPFN, but can leverage a broad class of predictive components.
>
> - **Expanded predictor suite.** In the revision, we expanded our evaluation to include CatBoost, LightGBM, and the recent MITRA foundation model, in addition to RF and XGBoost.
> - **Calibration.** As suggested, we applied Platt scaling to all standard estimators to improve their calibration before using them inside PEP.
> - **Results.** As shown in **Table 3 (Section 4)**, PEP generally improves over the CAM pruning baseline across these predictors and is never substantially worse. TabPFN still yields the highest absolute scores, which is consistent with its stronger calibration, but the *relative* benefit of PEP over marginal pruning is visible for a range of backbones.
>
> **[Action]** We added **Table 3** in **Section 4**, reporting results for multiple predictors with explicit calibration and discussing how PEP affects each of them.
>
> ---
>
> > **[W2]** *All synthetic data experiments are limited to $d=10$... The paper only reports runtime for $d=10$, where its runtime is already noticeably higher than CAM pruning. When the graph density increases... the pruning cost is at least $O(d^2)$.*
>
> This was a critical limitation of the initial submission. In the revision, we both improve the pruning algorithm and extend the experiments to larger graphs.
>
> - **Algorithmic change.** **Section 3.4** introduces a Hierarchical Group Pruning strategy that evaluates groups of parents and only refines groups whose joint evidence is close to or above the MDL gate, instead of testing parents one by one.
>
> - **Complexity.** As analyzed in **Appendix B.4**, under a sparse regime with at most $s$ true parents per node, this reduces the number of evidence evaluations per node from a naive $\Theta(P_j^2)$ to $O(s \log P_j)$, where $P_j$ is the number of admissible parents. This substantially alleviates the quadratic pruning bottleneck of the original scheme.
>
> - **Scalability experiments.** We added **Table 4**, which reports performance on graphs with $d \in \{30, 50, 100\}$. PEP consistently improves over the baseline pruning across these dimensions.
>
> - **Runtime.** We added Figure 6, which shows that while PEP has higher overhead at $d = 10$, its runtime grows more slowly than the baseline as $d$ increases. At $d = 100$, PEP is significantly faster than the standard CAM pruning in our setup (about $7.5\times$ speedup).
>
> **[Action]** We added **Section 3.4** (Hierarchical Group Pruning), Table 4 (results up to $d = 100$), and **Figure 6** (runtime scaling) to demonstrate that, in the sparse regimes we target, the quadratic pruning bottleneck is substantially mitigated in both theory and practice.

---

> ### Author Response · Authors · 2025-12-03
>
> > **[W3]** *A core argument... is replacing "manually-tuned thresholds". However, its MDL gate (Eq. 3) includes a fixed overhead $\kappa$... calibrated once for the entire study... This feels like just swapping one "manually-tuned $\alpha$" for another.*
>
> We agree that the fixed constant $\kappa$ in the initial submission made the MDL gate look heuristic. In the revision, we reworked the penalty term so that it is structurally adaptive and tied to the problem size.
>
> - **New formulation.** In Eq. (3) and Eq. (4) of the revision, we replace the fixed $\kappa$ with a structural complexity term
>   $\Omega(n, d) = \eta * \ln n * \ln d^2,$ so that the gate now depends on both sample size $n$ and graph dimension $d$.
>
> - **Justification.** This term is motivated by MDL and extended BIC principles: $\ln n$ plays the usual role of a sample-size penalty, while $\ln d^2$ reflects the quadratic growth of the candidate edge set. We derive the overall gate from a two-part code-length argument (identity, sparsity, structural penalty) in **Section 3.3** and **Appendix B**, rather than introducing $\kappa$ as a free constant.
>
> - **Sensitivity of the scale $\eta$.** We retain a single global scale $\eta$, but we no longer tune it per dataset. In **Figure 5**, we sweep $\eta$ over a broad range and observe a wide plateau: performance is stable around $\eta = 1$ across all reported graph sizes $d \in \{10, \dots, 100\}$. We therefore fix $\eta = 1$ in all reported experiments.
>
> **[Action]** We replaced the heuristic constant $\kappa$ with the structurally adaptive term $\Omega(n, d)$ in Section 3.3, added a code-length derivation in **Appendix B**, and included a sensitivity study for $\eta$ (**Figure 5**) to show that a single default choice works robustly in our benchmark settings.

---

> ### Author Response · Authors · 2025-12-04
>
> > **[Q1]** *Could the authors provide an experiment on scaling with the number of nodes $d$?*
>
> Yes. We extended the synthetic experiments to larger graphs with $d \in \{30, 50, 100\}$ nodes. The structural accuracy results in **Table 4 (Page 9)** and **Figure 5 (Page 10)** show that PEP consistently improves SHD, SID, and F1 over the baseline pruning for all graph sizes, and that these gains become larger as $d$ increases. In addition, **Figure 6 (Page 10)** reports runtime as a function of $d$, where Hierarchical Group Pruning yields approximately linear scaling and overtakes the cubic CAM baseline around $d \approx 40$.
>
> **[Action]** Added large scale experiments and analysis in **Section 4** (“Scalability to High-Dimensional Graphs”).
>
> ---
>
> > **[Q2]** *It is suggested that the authors add an experiment: when using XGBoost or RF as the predictor, add an extra "post-hoc calibration" step (e.g., Isotonic Regression or Platt Scaling).*
>
> We have incorporated this suggestion. All non TabPFN predictors (RF, XGBoost, CatBoost, LightGBM, MITRA) are now post processed with Platt scaling before computing the prequential evidence scores. The updated calibrated results are reported in **Table 3 (Page 9)** and described in more detail in **Appendix F**. This calibration step makes the probability outputs comparable across predictors and leads to more stable MDL decisions.
>
> **[Action]** Applied Platt scaling to all standard predictors in the new experiments reported in **Table 3** and **Appendix F**.
>
> ---
>
> > **[Q3]** *Algorithm 1 uses Greedy Backward Elimination. What is the reason for choosing this strategy? At the beginning... the context $S$ used to evaluate $\delta_{i \to j}$ contains many irrelevant variables...*
>
> We chose backward elimination in order to reliably capture **synergies** (for example XOR or collider structures) that are invisible to marginal or forward selection.
>
> * **Rationale.** As discussed in **Appendix B.3**, when $X_j = X_1 \oplus X_2$ the marginal signals $I(X_j; X_1)$ and $I(X_j; X_2)$ are zero, so a forward selection strategy would discard both parents. Backward elimination instead evaluates each edge $i \to j$ in the presence of all other candidate parents $S \setminus \{i\}$, which allows the conditional gain $\delta_{i \to j}(q; S)$ to recover $I(X_j; X_1 \mid X_2)$ and retain such synergistic edges.
>
> * **Handling irrelevant variables efficiently.** We share the concern that naively starting from a large context $S$ can be both noisy and expensive. This is precisely why we introduce *Hierarchical Group Pruning* in **Section 3.4**, and analyze it formally in **Appendix B.3 and B.4**. Candidates are first ranked by a lightweight marginal score, then tested in groups. Groups whose joint evidence falls below the MDL gate are discarded after a constant number of evaluations, reducing the number of evidence computations from $O(P_j^2)$ to $O(s \log P_j)$ in the sparse regime. This quickly removes blocks of irrelevant variables while preserving the backward elimination logic on groups that still contain true parents.
>
> **[Action]** Added **Appendix B.3** (“Rationale and Complexity of Backward Elimination”) and clarified the role of *Hierarchical Group Pruning* in **Section 3.4**.

---

### Author Response · Authors · 2025-12-04

We thank the reviewers for their constructive feedback. The revised manuscript incorporates these suggestions, with **major changes and new additions highlighted in $\textcolor{blue}{\text{blue}}$ in the revised PDF** (minor stylistic or contextual edits for flow are not highlighted).

Below is a summary of the main improvements relative to the original submission.

1. **Clearer formulation and notation**

   * Rewrote the paragraph on “Positioning relative to prior paradigms” to use simpler language and to describe PEP along three clear axes (evidence estimation, decision criterion, applicability) (**Section 2**).
   * Clarified the data generating setting and notation, now explicitly defining the SCM, the meaning of $\mathbb{E}[\cdot]$, and the sets $\text{Pred}_\pi(j)$ and $P_j$ (**Section 3** and **Algorithm 1**).
   * Clarified several previously informal quantities, including the definition of the linearity parameter $\rho_{\text{lin}}$ used in the robustness to functional form experiment (**Section 4.1, Figure 4, Appendix F**).

2. **Adaptive structural MDL gate and structural penalty**

   * Replaced the previous fixed overhead term in the MDL gate with an explicit structural penalty $\Omega(n,d) = \eta \ln n ,\ln d^2$, and explained its strength, confidence, and complexity interpretation (**Section 3.3, Eq. (3) and (4)**).
   * Added a schematic comparison of fixed versus adaptive gates that shows how $\tau_{\text{MDL}}(S,i)$ adapts with $(n, P_j, k)$ and why it remains stable under imperfect predictors (**Appendix B.1–B.2, Figure B.1**).
   * Introduced an ablation of the scaling factor $\eta$ over $d \in {10, 30, 50, 100}$, confirming the theoretically motivated choice $\eta = 1.0$ as a robust default (**Section 4.1, Figure 5, Table G.1**).

3. **Hierarchical group pruning and scalability analysis**

   * Introduced a *Hierarchical Group Pruning* variant that applies the same PEP decision rule to groups of parents, reducing the number of evidence evaluations per node from $\Theta(P_j^2)$ to $O(s \log P_j)$ under sparsity (**Section 3.4, Appendix B.3–B.4**).
   * Added a detailed analysis of standard backward elimination, explaining why it is needed for synergy detection and quantifying its quadratic cost, before showing how the hierarchical variant mitigates this bottleneck (**Appendix B.3–B.4**).
   * Added new runtime experiments demonstrating that hierarchical pruning yields quasi linear scaling and up to a $7.5\times$ speedup over CAM pruning at $d = 100$ nodes (**Section 4.1, Figure 6**).

4. **Stronger empirical validation and isolating the role of TabPFN**

   * Expanded the robustness to predictive component study by instantiating PEP with RF, XGBoost, CatBoost, LightGBM, MITRA, and TabPFN. **Table 3** shows consistent improvements over CAM pruning for all predictors, indicating that the gains are not specific to TabPFN (**Section 4.1, Table 3**).
   * Added new experiments on larger graphs with $d = 30, 50, 100$ across several ordering backbones. **Table 4** shows that PEP’s advantage over standard pruning becomes more pronounced as $d$ increases, supporting the design of the structural penalty and addressing concerns about scalability beyond $d = 10$ (**Section 4.1, Table 4**).

Reviewer `Mmos` and `E4Pd` highlighted as strengths that our work isolates the pruning step as a key bottleneck in ordering based causal discovery, introduces a clear information theoretic “Panoramic view” alternative to traditional marginal pruning, and already shows strong empirical gains and robustness stress tests.

Building on these acknowledged strengths, the revision sharpens the theory (explicit CMI target, finite sample guarantees, and a structural MDL gate), clarifies the exposition, and adds extensive robustness and scalability evidence, including misspecification stress tests, mixed functional forms, different predictive components, random orderings, and large scale graphs.

Taken together, the acknowledged strengths and these new robustness and scalability results support PEP as a mature and practically useful reframing of pruning, and we hope the paper can now be considered a strong candidate for acceptance at ICLR 2026.

---

### Meta-Review · Area_Chair_i5Au · 2026-01-08

**Summary:**

The paper aims to reframe pruning in ordering-based causal discovery as a local information-theoretic model selection problem, retaining an edge only when its out-of-sample prequential log-likelihood gain exceeds an adaptive MDL penalty, with the population target equal to conditional mutual information.

**Reviewer Concerns:**

- clarity of exposition needs improvment; the current version hinders understanding; notation/definitions introduced too late, making the theory hard to follow.

- ambiguous probabilistic/statistical setup; expectations/probabilities not clearly stated w.r.t. which distribution

- Theoretical statements feel underspecified

- Figures/metrics in empirical studies hard to interpret

- empirical gains appear dependent on TabPFN; with RF/XGBoost results are weaker or sometimes worse; suggests reliance on a highly calibrated density estimator

- lack of scalability and potentially high computational cost; Experiments mostly limited to small graphs (10–20 nodes); lacks evidence for 100–200+ variables.

**Reviewer Scores:**

The reviewers were largely negative about the paper, and the revised manuscript improved in exposition but involves substantial revision that would potentially require another round of thorough review; the discussion may not have changed the situation.

---

### Decision · Program_Chairs · 2026-01-26

Reject